# Discrete Factorial Representations as an Abstraction for Goal Conditioned RL

**Riashat Islam**[1,5,7*]**, Hongyu Zang**[3]**, Anirudh Goyal**[2,7,8]**, Alex Lamb**[6,7]**,**
**Kenji Kawaguchi**[4]**, Xin Li**[3]**, Romain Laroche**[5]**, Yoshua Bengio**[2,7]**, Remi Tachet Des Combes**[5]
[1] McGill University, Montreal, Canada, [2] University of Montreal, Montreal, Canada
[3] Beijing Institute of Technology, Beijing, China, [4] Harvard University, Cambridge, USA
[5] Microsoft Research, Montreal, Canada, [6] Microsoft Research, New York, USA
[7] Mila - Quebec AI Institute, Montreal, Canada [8] DeepMind

## Abstract

Goal-conditioned reinforcement learning (RL) is a promising direction for training agents that are capable of solving multiple tasks and reach a diverse set of objectives. How to *specify* and *ground* these goals in such a way that we can both reliably reach goals during training as well as generalize to new goals during evaluation remains an open area of research. Defining goals in the space of noisy and high-dimensional sensory inputs poses a challenge for training goal-conditioned agents, or even for generalization to novel goals. We propose to address this by learning factorial representations of goals and processing the resulting representation via a discretization bottleneck, for coarser goal specification, through an approach we call DGRL. We show that applying a discretizing bottleneck can improve performance in goal-conditioned RL setups, by experimentally evaluating this method on tasks ranging from maze environments to complex robotic navigation and manipulation. Additionally, we prove a theorem lower-bounding the expected return on out-of-distribution goals, while still allowing for specifying goals with expressive combinatorial structure.

## 1 Introduction

Reinforcement Learning is a popular and highly general framework [25, 60] focusing on how to select actions for an agent to yield high long-term sum of rewards. An important question is how to control the desired behavior of an RL agent, both during training and evaluation [24]. One way to control this behavior is by specifying a reward signal [52, 56]. While this approach is very general, the reward signal can be hard to design and may not be the most informative form of feedback. The credit assignment problem in RL can become difficult when the reward signal is sparse [62, 63, 36, 37, 57], such as policy gradients becoming nearly flat in regions where reward is almost never achieved. Generalization can also suffer if the agent only learns one way to achieve a high reward rather than learning a diverse set of skills for coping with novel challenges [20].

One potential way to flexibly specify and ground the desired behavior of RL agents is by training agents that receive a reward when they reach a goal specified explicitly to them [23]. In this approach, called *Goal-Conditioned RL*, a single agent is trained to reach a diverse set of goals, and is given a reward only when it reaches the goal it was instructed to reach [61, 48, 44]. This provides a richer signal for the agent than simply collecting more samples oriented around a single goal, as reaching multiple goals requires the agent to learn a more diverse and robust set of skills. It also allows for more flexible and tightly constrained control over its desired behavior [12, 5, 27]. Finally, the diversity of goals seen during training should help improve both credit assignment and generalization [48, 43].

---

*Corresponding Author. E-mail: riashat.islam@mail.mcgill.ca, anirudhgoyal9119@gmail.com

36th Conference on Neural Information Processing Systems (NeurIPS 2022).

While this framework is promising, it introduces two new challenges: *goal grounding* [8, 2] and *goal specification* [4]. Goal grounding refers to defining the goal space, and goal specification refers to selecting what goal the agent should try to reach in a given context. The agent is only rewarded in goal-conditioned RL when reaching the reward it was instructed to reach, whereas in goal-free RL a reward is provided regardless of any such specification, which makes the nature of the agent's task fundamentally different.

What makes grounding and specifying goals challenging? Consider trying to train a goal-conditioned RL agent to pick up various fruits from a table. For example, we may want it to pick up a red apple or a green pear (illustrated in Figure 1). The number of possible goals of interests may be fairly small, such as the set of all valid combinations of fruits and their colors, while the number of possible observations of goals is extremely large when working in a rich observation space (*e.g* images from a camera). *Goal Grounding* refers to this challenge of relating high-dimensional observations and the space of relevant goals. *Goal Specification* refers to picking a suitable goal for the

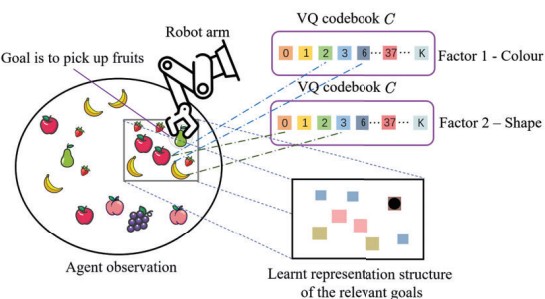

Figure 1: Illustration of learning *discrete* and *factorial* goal representations.

agent to reach and computing an appropriate reward when it is reached. It also implies specifying goals reachable in the agent's current context [39, 26]. Goal specification can be done either manually by a developer or by another RL agent, such as a high-level agent which generates goals a lower-level agent then tries to reach [12, 5, 27, 22]. Goals specified in language are an excellent fit for these desiderata, as language is a compressed discrete representation which is useful for out-of-distribution generalization, while being compositional and expressive [19, 11, 22, 17, 1, 67]. At the same time, connecting language feedback for an agent is non-trivial, requiring special assumptions or a labeling framework [9].

We propose to learn the goal representations with self-supervised learning (either trained on their own, or jointly with the downstream RL objective) while forcing them to be *discrete* and *factorial*. To perform this discretization, we use Vector-Quantization [64, 47, 33] which discretizes a continuous representation using a codebook of discrete and learnable codes. The approach proposed here, called DGRL, serves two complementary purposes. First, it provides a structured representation of the raw visual goals. By representing the visual goals as a *composition* of discrete codes from a learned dictionary, it simplifies the grounding of unseen goals, *i.e.*, goals not seen during training, to novel compositions of the trained discrete codes. We show empirically that this improves the generalization performance of goal-reaching policies while remaining expressive enough. Second, the learned discrete codes can be used by another agent (like a higher-level policy in hierarchical RL) to specify sub-goals to a lower-level policy, and eventually complete the task (*i.e.*, reach the final goal). In this case, goal-inference is learned end-to-end. The effectiveness of goal-conditioned HRL relies on the specification of semantically meaningful sub-goals. Using factorial discrete sub-goals allows the higher-level policy to specify semantically meaningful objectives to the lower-level policy.

## 2 Preliminaries

**Goal-conditioned RL.** We consider a goal-conditioned Markov Decision Process, where the goals $g \in \mathcal{G}$ live in the state space $\mathcal{S}$, *i.e.*, $\mathcal{G} = \mathcal{S}$. We denote a goal-conditioned policy as $\pi(a|s, g)$ (either stochastic or deterministic), and its expected total return as $J(\pi) = \mathbb{E}\left[ \sum_{t=0}^{T} R(s_t, g, a) \right]$ where the goal $g$ is either sampled from a distribution $\rho_g$ or provided by another higher level policy $\pi_{\theta_h}^h(g|s)$. The value function $V^\pi$ is additionally conditioned on goals, and is trained to predict the expected sum of future rewards conditioned on states and goals; $V^\pi(s, g) = \mathbb{E}\left[ \sum_{t=0}^{T} R(s_t, g, a) \mid s_0 = s; \pi \right]$. As in standard RL, the objective in goal-conditioned RL is to maximize the expected discounted returns induced by the goal-conditioned policy.

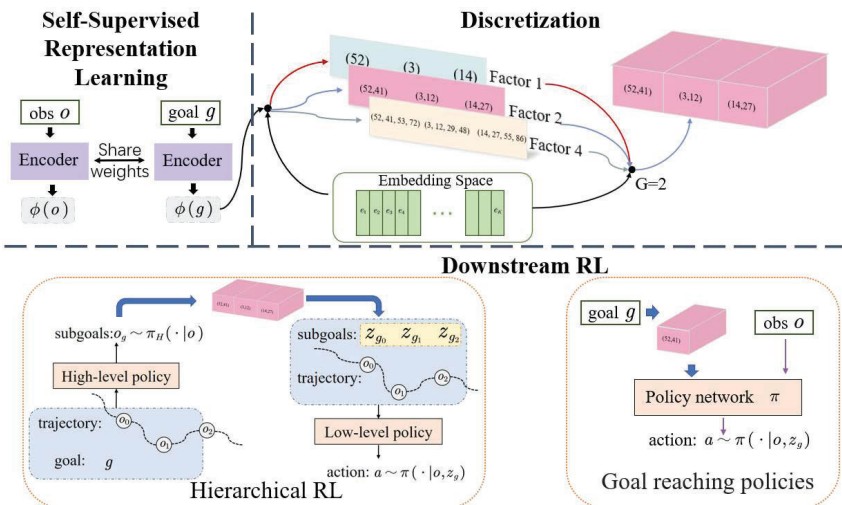

Figure 2: **Summary of Proposed DGRL Model** for improving *goal grounding* and *goal specification* by making goal representations *discrete* and *factorial*. We learn a latent representation for both observations and goals using a self-supervised learning method (sec. 3). We convert the learnt latent representation into discrete latents based on a VQ-VAE quantization bottleneck with multiple factor outputs (sec. 3). We use the resulting discrete representations for downstream RL tasks: (i) to train a goal-conditioned policy or value function, and (ii) in the context of goal-conditioned hierarchical reinforcement learning (sec. 3.1).

**Hierarchical Reinforcement Learning.** We consider goal-conditioned settings in which the goals are specified in the observation space. In the hierarchical reinforcement learning (HRL) setup, goals are provided by a higher level policy $\pi_{\theta_h}^h(g|s_t)$. The higher level policy operates at a coarser time scale and chooses a goal $g_t \sim \pi_{\theta_h}^h(g|s_t)$ to reach for the lower level policy every K steps. The lower level policy executes primitive actions $\pi_{\theta_l}^l(a|s_t, g_t)$ to reach the goals specified by the high-level policy and is trained to maximize the intrinsic reward provided by the high-level policy. The higher level policy is trained to maximize the external reward *i.e.*, the reward function specified by the MDP. Both the higher and lower level policies can be trained with any standard RL algorithms, such as Deep Q-Learning (DQN) [36] or policy optimization based algorithms [50, 51, 30]. Alternately, one can also consider another setup for goal-conditioned RL, where the goals are provided by the environment $g_1, \ldots g_L$, and are part of the state or observation space. At each episode of training, one of the goals is sampled from the distribution of goals $\rho_g$ and the policy is trained to reach the sampled goal. At test time, the agent can be evaluated either on its ability to reach goals within the distribution $\rho_g$, or for its out-of-distribution generalization capability to reach new kinds of goals. We consider both the HRL and goal-conditioned setups, and evaluate the significance of learning factorial representation of discrete latent goals in a series of complex goal-conditioned tasks.

**Vector Quantized Representations.** VQ-VAE [64, 47, 33] discretizes the bottleneck representation of an auto-encoder by adding a codebook of discrete learnable codes. The input is passed through an encoder. The output of the encoder is compared to all the vectors in the codebook, and the codebook vector closest to the continuous encoded representation is fed to the decoder. The decoder is then tasked with reconstructing the input from this quantized vector.

**Self-supervised learning of representations.** Several papers [31, 58, 53, 34] have demonstrated the benefits of using a pre-training stage where the representations of raw states are learned using self-supervised objectives in a task-agnostic fashion. After the pre-training stage, the representations can be used for (and potentially also fine-tuned on) downstream tasks. These self-supervised representations have been shown to improve sample efficiency.

## 3 Discrete Goal-Conditioned Reinforcement Learning (DGRL)

In this section, we provide technical details on the proposed framework, DGRL, which consists of three parts: (a) learning representations of raw visual observations through self-supervised representation objectives, (b) processing the resulting representations via a learned dictionary of discrete

codes, and (c) using the resulting discrete representations for downstream goal-conditioned and HRL tasks. We later describe, in Section 6, how discrete goal representations can accelerate learning in complex navigation and manipulation tasks. We emphasize that these representations can be learned at the same time as the downstream-RL objective or pre-trained with self-supervised learning, and then used as a fixed representation for RL.

**Self-Supervised Goal Representation Learning.** One can use any off-the shelf self-supervised method for learning representations of the raw state and the goal observations. We denote by $\phi$ the encoder network that takes as input the raw state and maps it to a continuous embedding: $z_e = \phi(s)$. Here, we explore two different self-supervised techniques for learning representations. For simpler environments, we use a simple autoencoder with its standard reconstruction objective. For more complex environments, we use the Deep InfoMax approach [34] which optimizes for a contrastive objective as a proxy to maximizing the mutual information between representations of nearby states in the same trajectory.

**Processing continuous representations via a discrete codebook.** We learn discrete representations by using the vector-quantization method from the VQ-VAE paper [64], and follow the multi-factor setup used in Discrete-Value Neural Communication [33]. The discretization process for each vector $z_e \in \mathcal{H} \subset \mathbb{R}^m$ is described as follows. First, vector $z_e$ is divided into $G$ segments $c_1, c_2, \ldots, c_G$, where $z_e = \text{CONCATENATE}(c_1, c_2, \ldots, c_G)$, and each segment $c_i \in \mathbb{R}^{m/G}$ (implying that $m$ is divisible by $G$). Each continuous segment $c_i$ is mapped independently to a discretized latent vector $e \in \mathbb{R}^{L \times (m/G)}$ where $L$ is the size of the discrete latent space (*i.e.*, an $L$-way categorical variable):

$$e_{o_i} = \text{DISCRETIZE}(c_i), \quad \text{where } o_i = \underset{j \in \{1, \ldots, L\}}{\arg \min} \, ||c_i - e_j||.$$

These discrete codes, which we call the factors of the continuous representation $z_e$, are concatenated to obtain the final discretized vector $z_q$:

$$z_q = \text{CONCATENATE}(\text{DISCRETIZE}(c_1), \text{DISCRETIZE}(c_2), ..., \text{DISCRETIZE}(c_G)). \quad (1)$$

The loss for vector quantization is: $\mathcal{L}_{\text{discretization}} = \frac{\beta}{G} \sum_i^G ||c_i - \text{sg}(e_{o_i})||_2^2$.

The training procedure closely follows both [33] and [64]. Here, sg refers to a stop-gradient operation that blocks gradients from flowing into $e_{o_i}$, and $\beta$ is a hyperparameter which controls how strongly we move the codes toward the encoded values. Unlike [33], we used a moving average to update the code embeddings rather than learning them directly as parameters. We update $e_{o_i}$ with an exponential moving average to encourage it to become close to the selected output segment $c_i$. This update sets the new value of $e_{o_i}$ to be equal to $\eta e_{o_i} + (1 - \eta)c_i$, where the value of $\eta$ is a fixed hyper-parameter controlling how quickly the moving average updates.[2] The term $\sum_i^G ||c_i - \text{sg}(e_{o_i})||_2^2$ is often called the commitment loss. We trained the VQ-quantization process together with other parts of the model by gradient descent. When there were multiple $z_e$ vectors to discretize in a model, the mean of the commitment loss across all $z_e$ vectors was used.

**Summary.** The multiple steps described above can be summarized by $z_q = q(z_e, L, G)$, where $L$ is the codebook size, $G$ the number of factors per vector, and $q(\cdot)$ the whole discretization process. We train the representations for both the state and goal observations with this discretization bottleneck applied to the continuous representations resulting from the self-supervised training. The number of factors $G$ is a hyper-parameter. In our experiments, we explored different values: $G = 1, 2, 4, 8, 16$, and found that $G = 16$ worked the best. Discretizing with more factors slightly increases computation but reduces the number of model parameters due to the codebook embeddings being reused across the different factors.

## 3.1 Using representations for downstream RL

We use the discrete representations for downstream RL tasks: (i) to train a goal-conditioned policy, and (ii) in the context of hierarchical reinforcement learning.

**Goal-conditioned RL.** Defining goals in the space of noisy, high-dimensional sensory inputs poses a challenge for generalization to novel goals because the encoder that maps the goal observations to

---

[2]Note that this could also be thought of as a gradient step on $e_{o_i}$ taken in the direction $c_i - e_{o_i}$.

the low dimensional latent representation may fail to generalize. One way to address this is to embed the continuous latent representation into a discrete representation such that the representation of the novel goal is mapped to the fixed set of latent discrete codes. This facilitate generalization to new combinations of these codes while making it easy for downstream learning to figure out the meaning of each discrete code. In this setup, instead of feeding the continuous state and goal embeddings to the agent, we use their discretized versions, thus grounding goal representations in the input space.

We use the resulting representations to train a goal-conditioned policy $a_t \sim \pi^l_{\theta_l}(a|s_t, g_t)$ or a goal-conditioned action value function $Q(s_t, a_t, g_t)$. At each training episode, a goal is sampled from the goal distribution $\rho_g$, and the agent gets rewarded for reaching it. This reward can either be *extrinsic*, *i.e.*, part of the environment, or *intrinsic*, *i.e.*, part of the algorithm. In DGRL, we define the intrinsic reward as the fraction of discrete factors which match in the respective representations of the goal observation and of the state observation. At test time, the agent can either be evaluated on reaching goals within the distribution $\rho_g$, or for its generalization capability to goals not seen during training.

**Hierarchical RL.** The higher level policy $g_t \sim \pi^h_{\theta_h}(g \mid s_t)$ outputs a continuous representation of goals $g$ by conditioning on the states every $K$ time-steps, it can also output a sub-goal $s_g$ by conditioning on both states $s$ and environment goals $g$, *i.e.*, $\pi^h_{\theta_h}(s_g \mid s_t, g_t)$. The effectiveness of goal-conditioned HRL relies on the specification of semantically meaningful sub-goals. Learned codebooks (Section 3) consisting of a set of discrete codes can be used by a higher level policy to *specify* which goal to reach to a lower level policy. The use of learned codebooks ensures that the goal specified by the higher level policy is grounded in the space of raw-observations.

In Section 6, we empirically show the benefits of the proposed approach for training goal-reaching policies or goal-conditioned value functions, as well as in a goal-conditioned hierarchical RL setup.

## 4    Theoretical Analysis

In this section, discretization is shown to improve generalization to novel goals by enhancing the concentration of the goal distribution within each neighborhood of discretized goal values; *i.e.*, by decomposing the goal probability $p(g)$ into $p(g) = \sum_k p(g|g \in \mathcal{G}_k)p(g \in \mathcal{G}_k)$ with the neighborhood set $\{\mathcal{G}_k\}_k$, it improves the overall performance in $p(g)$ by increasing the concentration in $p(g|g \in \mathcal{G}_k)$. Intuitively, this is because the discretization removes varieties of possible goal values $g \in \mathcal{G}_k$ for each neighborhood $\mathcal{G}_k$. To state our result, we define $\varphi_\theta(g) = \mathbb{E}_{s_0}[V^\pi(s_0, g)]$, where $\theta \in \mathbb{R}^m$ is the vector containing model parameters learned through $n$ goals observed during training phase, $g_1, \ldots, g_n$. We denote the discretization of $g$ by $q(g)$, and the identity function by id as $\mathrm{id}(g) = g$. Let $\mathcal{Q} = \{q(g) : g \in \mathcal{G}\}$ and $\hat{d}$ be a distance function. We use $\mathcal{Q}_i$ to denote the $i$-th element of $\mathcal{Q}$ (by ordering elements of $\mathcal{Q}$ with an arbitrary ordering). We also define $[n] = \{1, \ldots, n\}$, $\mathcal{G}_k = \{g \in \mathcal{G} : k = \arg\min_{i \in [|\mathcal{Q}|]} \hat{d}(q(g), \mathcal{Q}_i)\}$, $\mathcal{I}_k = \{i \in [n] : g_i \in \mathcal{G}_k\}$, and $\mathcal{I}_Q = \{k \in [|\mathcal{Q}|] : |\mathcal{I}_k| \geq 1\}$. We denote by $c$ a constant in $(n, \theta, \Theta, \delta, S)$.

The following theorem (proof in Appendix D) shows that the goal discretization improves the lower bound of the expected sum of rewards for unseen goals $\mathbb{E}_{g \sim \rho_g}[(\varphi_\theta \circ \varsigma)(g))]$ by the margin of $\omega(\theta)$:

**Theorem 1.** *For any $\delta > 0$, with probability at least $1 - \delta$, the following holds for any $\theta \in \mathbb{R}^m$ and $\varsigma \in \{\mathrm{id}, q\}$:*

$$\mathbb{E}_{g \sim \rho_g}[(\varphi_\theta \circ \varsigma)(g))] \geq \frac{1}{n} \sum_{i=1}^n (\varphi_\theta \circ \varsigma)(g_i) - c\sqrt{\frac{2\ln(2/\delta)}{n}} - \mathbb{1}\{\varsigma = \mathrm{id}\}\omega(\theta)$$

*where* $\omega(\theta) = \frac{1}{n} \sum_{k \in \mathcal{I}_Q} |\mathcal{I}_k| \left(\frac{1}{|\mathcal{I}_k|} \sum_{i \in \mathcal{I}_k} \varphi_\theta(g_i) - \mathbb{E}_{g \sim \rho_g}[\varphi_\theta(g)|g \in \mathcal{G}_k]\right)$. *Moreover, for any compact $\Theta \subset \mathbb{R}^m$, if $\varphi_\theta(g)$ is continuous at each $\theta \in \Theta$ for almost all $g$ and is dominated by a function $\chi$ as $|\varphi_\theta(g)| \leq \chi(g)$ for all $\theta \in \Theta$ with $\mathbb{E}_g[\chi(g)] < \infty$, then the following holds:*

$$\sup_{\theta \in \Theta} |\omega(\theta)| \xrightarrow{P} 0 \quad when \quad n \to \infty.$$

*Proof.* Detailed proof provided in the Appendix D ☐

Without the goal discretization, we incur an extra cost of $\omega(\theta)$, which is expected to be strictly positive since $\frac{1}{|\mathcal{I}_k|} \sum_{i \in \mathcal{I}_k} \varphi_\theta(g_i)$ is maximized during training while $\mathbb{E}_{g \sim \rho_g}[\varphi_\theta(g)|g \in \mathcal{G}_k]$ is not.

Thus, the goal discretization can improve the expected sum of rewards for unseen goals by the degree of $\omega(\theta)$, which measures the concentration of the goal distribution in each neighborhood. This extra cost $\omega(\theta)$ goes to zero when the number of goal observations $n$ approaches infinity.

## 5 Related Work

Learning with multiple hierarchies has long been proposed in the RL literature, where goal conditioned HRL implements high level planning and low level control using sub-goals. Often in goal conditioned HRL, the higher level policy specifies goals which may not have good *specification* and *grounding*. Several prior works focus on goal-conditioned RL to improve sample efficiency in deep RL tasks [38, 40]. These build on ideas that were proposed years back to solve long horizon tasks by hierarchical RL specifying goals [23, 13, 14, 68]. The goal is to learn to solve sub-goals provided to the policy, by learning to predict a sequence of actions that can reach each of the sub-goals [66, 49, 41]. Additionally, in existing HRL literature, distance measures are often used based on goal-conditioned value functions, allowing to measure distances between states and the sequence of sub-goals to reach [16, 71], for planning [42], or exploration [29]. Since the set of goals specified in the state space can be arbitrary, an additional constraint is often also learnt to tie the distribution of selected goals to those the lower level policy can reach [72]. We tackle this problem by proposing DGRL, for better *grounding* and *specification* of sub-goal representations.

In previous works, mutual information based objectives have been proposed for goal conditioned RL. They perform goal-based representation learning, in order to improve stability of training goal-conditioned value functions [38, 40], or to provide goal representations allowing to identify decision states for better exploration [18]. However, for most of these settings, the sub-goals are based on an external reward and are lacking in terms of specification, which can lead to inefficient training. [32, 72] have proposed approaches that penalize the high level controller for generating sub-goals that are too difficult for the lower level policies, through the use of additional constrained objectives [72]. We highlight that our proposed DGRL can be generically applied to any goal conditioned RL literature for better *grounding* and *specification* of the sub-goals. Furthermore, recent work has shown significance of learning representations through self supervised objectives in RL [3, 54], often as a pre-training phase [55, 69], which can help for both exploration [35] and control [70].

In the context of goal conditioned RL, it can be a challenging problem as it additionally requires learning reliable representations of goals in parallel, purely from high dimensional observations [15]. Previous works [32] have often used the entire observation space as goals, which is not scalable for complex tasks. Other works have used a pre-defined space of sub-goals as domain knowledge [38], or self-play for sub-goal representations [59] to reduce the complexity of goal space design. Most recently, [42, 41] utilized unsupervised representation learning to learn a goal representation space, which can further be used for planning and control. In this work, we show additionally that using a bottleneck can further lead to factorial representation of goals, while helping with goal specification via learning a latent space of discrete goals usable for planning and control. We emphasize that DGRL can be integrated on any existing goal conditioned approach that utilizes learning a sub-goal representation.

## 6 Experiments

The main goal of our experiments is to show that goal discretization can lead to sample efficient learning and generalization to novel goals, in goal-conditioned RL. First, we directly study this by training on environments with a set of goals (such as 8 positions within a gridworld) and then evaluating the agent's ability to reach a position within the gridworld which it was not trained to reach. Second, we consider hierarchical goal-conditioned RL, in which a higher-level agent generates goals that a lower-level agent is tasked with reaching. In this case, the task of reaching novel goals occurs *organically* as the higher-level model selects new goals. This setup also shows the advantages of DGRL for *goal specification*. A secondary goal of our experiments is to show that using many discrete factors is often critical for optimal performance, which proves the value of *factorization* in *grounding goals*.

We evaluate our proposed method DGRL by integrating it into existing state-of-the-art goal-conditioned and hierarchical RL tasks. Experimentally, we analyse DGRL on several challeng-

ing testbeds that have previously been used in the RL community. DGRL in principle can be applied to any existing downstream goal-conditioned RL tasks. We demonstrate improvements on five such tasks. We consider maze navigation where images are used as observations and we show improved generalization to novel goals. We integrate DGRL to an existing goal-conditioned baseline for navigating procedurally-generated hard exploration Minigrid environments [10] and find that it outperforms state-of-the-art exploration baselines. We also show improvements with DGRL on continuous control (Ant) navigation and manipulation tasks, where goals come from a high-level controller. Finally, we show that discrete representations also significantly improve sample efficient learning on a challenging vision-based robotic manipulation environment.

**Demonstrating Factorized Representation Learning** We first pick a color-mnist supervised learning example to support the idea that DGRL can learn factorized or compositional representations. Figure 3 displays reconstructed images from a trained decoder operating on a discretized 2-factor

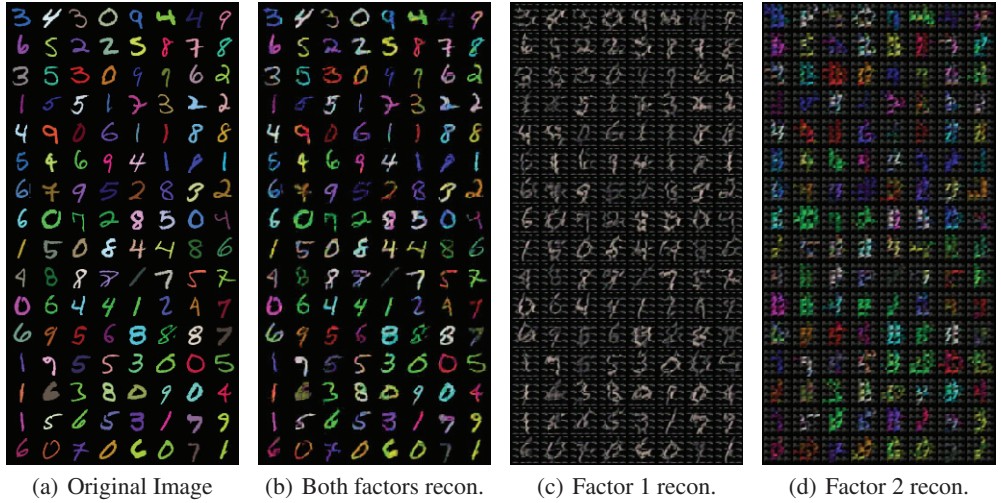

| (a) Original Image | (b) Both factors recon. | (c) Factor 1 recon. | (d) Factor 2 recon. |

Figure 3: Color-MNIST example to demonstrate factorized representations; reconstructing the original images with two factors. Leftmost : Original Image; Left-Middle : Reconstructed Image without substitution; Right-Middle : Reconstructed Image with one groups of discrete codes substituted by zero vectors; Rightmost : Reconstructed Image with the other groups of discrete codes substituted by zero vectors

representation. We find that different factors capture information of different semantic nature. More precisely, factor 1 tends to encode the shape of the digit, while factor 2 specialized in its color. This empirically suggests the emergence of "factorization" in the learnt representations. Further experimental details are provided in section C.1.

**Learning to Reach Diverse and Novel Goals.** We study a gridworld navigation task in which an agent is trained to reach a goal from a small finite set of training goals, and during evaluation is tasked with reaching a novel goal unseen during training. This is a navigation task with a pixel-level observation space showing the position of the agent and the goal in a gridworld. We consider two mazes spiral and single-loop topology. Experiment setup is given in Appendix C.2.

For this task, we train a goal-conditioned Deep Q-Learning (DQN) agent, and use a pre-trained representation $\phi(\cdot)$ where the encoder is trained using data from a random rollout policy. Because the gridworld is small the random rollout policy achieves good coverage of the state space, so we found this was sufficient for learning a good goal representation. At each episode, a specific goal is randomly sampled from a distribution of goals, and the DQN agent is trained to reach the specified goal for that episode. During evaluation, we test the learned agent on goals either from the training distribution, or not seen during training.

Furthermore, for this task, we additionally use an intrinsic reward to promote exploration of the goal-DQN agent. Since we learn a discrete factorial representation of the goal, we compute an exploration bonus based on the discrete latent codebooks; *i.e.*, we embed the states and goals using

the learned codes and then compute an intrinsic exploration bonus based on the fraction of learned factors that match. For the baseline goal-DQN agent, we provide an additional reward bonus based on the cosine distance between continuous embeddings of the state observation and goal. Figure 4 shows that DGRL significantly outperforms a continuous baseline goal DQN agent, when trained on either four goals or eight goals. We evaluate generalization to 4 novel goals unseen during training (Figure 5) and demonstrate improved generalization.

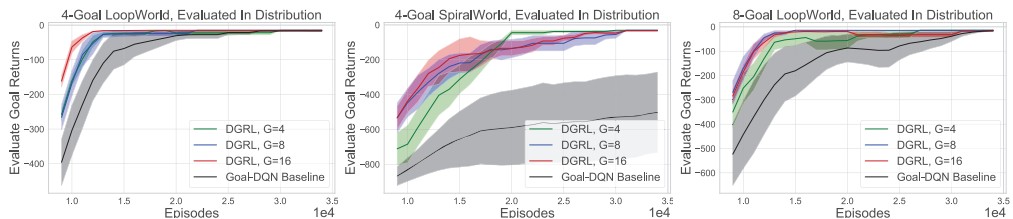

Figure 4: Loopworld maze environment. We show that for $4, 8$ and $16$ different discrete factors, DGRL outperforms a goal-DQN baseline agent with continuous goal representations. As we increase the number of factors $G$ to 16, the expressivity of the discrete goal representation increases, lowering the odds of the factors being the same. This provides a better intrinsic reward signal for exploration, resulting in faster convergence for DGRL integrated on a goal-DQN agent.

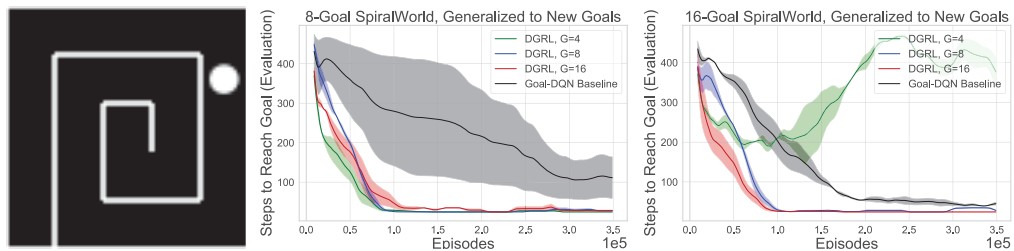

Figure 5: SpiralWorld environment (left). Generalization to a test distribution containing 4-goals in a SpiralWorld environment (left). We show the total number of steps to solve all test set goals, when trained on either an 8-goal or 16-goal training distribution.

In the previous experiment, we evaluated the generalization ability of DGRL by showing that learning discrete factorial representations of goals can improve generalization to novel goals. Now, we consider various setups in which a goal generating agent specifies goals using the learned codebook and a goal-conditioned agent is tasked with reaching the goals specified by the goal generating agent. We test various settings, where the goal generating agents is parameterized as an adversarial teacher [6], or as a higher-level policy in the case of hierarchical RL.

**Procedurally Generated MiniGrid Exploration Task.** We follow the experimental setup of [6] and [46] and evaluate DGRL on procedurally generated MiniGrid environments [10]. In [6], a goal-generating teacher proposes goals to train a goal-conditioned "student" policy. We integrate DGRL on top of AMIGO [6] and compare DGRL on a hard exploration task with state-of-the-art exploration baselines. Experimental results are summarized in Table 1 and more details provided in Appendix C.4. Note that unlike RIDE and RND, we do not provide an additional exploration bonus to DGRL, and find that DGRL can still solve this hard exploration task more efficiently.

**Goal Grounding in KeyChest Maze Navigation Domain.** We consider a simple discrete state action KeyChest maze navigation task, following [72], where discrete goals in the state space are provided by a higher level policy. For this task, to integrate DGRL, we learn an embedding $\phi(\cdot)$ of the goals, then discretize the representation with a learned codebook. We compare with a baseline HRAC [72] agent (details in Appendix C.3). Figure 7 shows an illustration of the KeyChest environment and a performance comparison of DGRL with different group factors $G$. Using fewer factors ($G = 4$) performs worse than the HRAC baseline, whereas using a larger number of factors ($G = 8$ or $G = 16$) improves the sample efficiency of the goal reaching agent, providing evidence for the benefits of factorization.

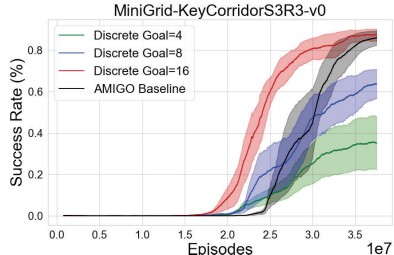

| Model | KCmedium |
|---|---|
| AMIGO + DGRL, G=16 | .96 ± .01 |
| AMIGO + DGRL, G=8 | .70 ± .16 |
| AMIGO | .93 ± .06 |
| RIDE | .90 ± .00 |
| RND | .89 ± .00 |
| ICM | .42 ± .21 |

Figure 6: Performance comparison of the Amigo baseline [6, adversarially intrinsic goals] with and without DGRL for goal discretization.

Table 1: We added DGRL on top of the Amigo baseline implementation provided by the authors.

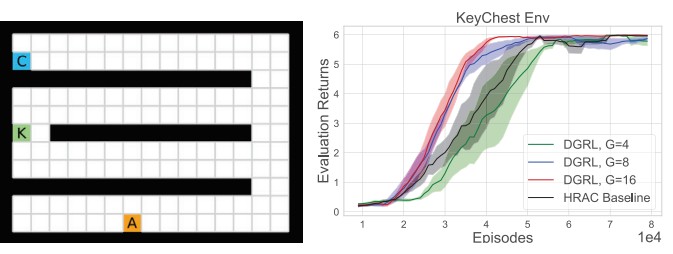

Figure 7: In KeyChest, the agent (A) starts from a random stochastic position, picks up the key (K) and then uses the key to open the chest (C). We find DGRL improves sample efficiency over the HRAC baseline.

**Ant Manipulation Control Domains.** We employed DGRL on three different continuous control tasks: AntMazeSparse, AntFall and AntPush. We emphasize that these tasks are the more challenging counterparts of AntGather and AntMaze tasks, typically used in the hierarchical RL community [38, 40]. Figure 8 provides an illustration. We evaluate goal discretization by integrating DGRL to the state-of-the-art HRAC baseline. Details of the experimental setup are provided in Appendix 6. Figure 8 shows that specifying the goals using the learned codebook helps DGRL achieve a higher success rate compared to the HRAC baseline.

**Ant Navigation Maze Tasks.** We consider Ant navigation tasks that require extended temporal reasoning, following the setup in Reinforcement learning with Imagined Subgoals [7, RIS]: a U-shaped maze, and an S-shaped maze (the S-shaped maze is shown in Figure 9). The ant navigating in the maze is trained to reach any goal in the environment. The agent is evaluated for generalization in an extended temporal setting with a difficult configuration, we compare the success rate of DGRL integrated on top of RIS with several baselines. We emphasize the difficulty of these tasks, where existing baselines like soft actor critic [20, SAC] and temporal difference models [45, TDM] fail completely. Results in Figure 9 show that DGRL improves the sample efficiency over the RIS baseline. Additional experimental setup and environment configurations are provided in Appendix C.6.

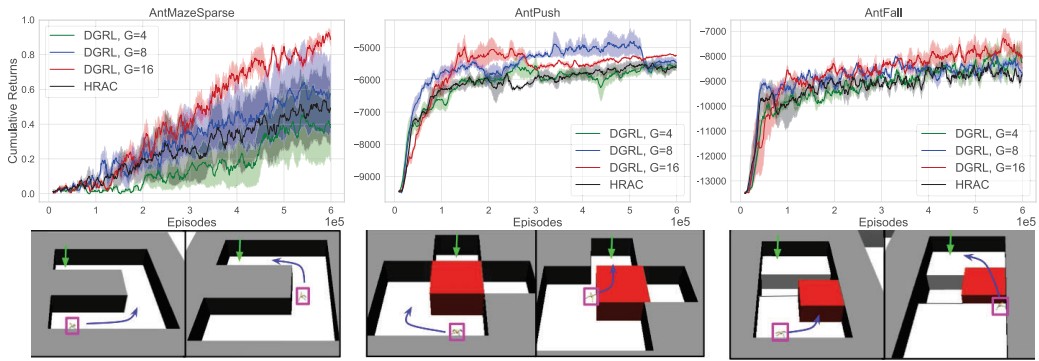

Figure 8: Comparison of DGRL with baseline HRAC [72] on 3 different navigation tasks.

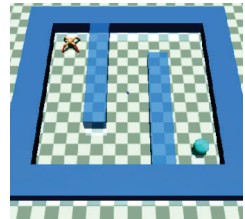 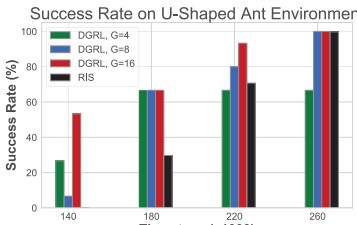 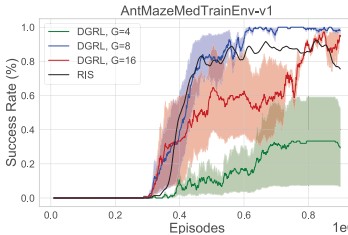

Figure 9: Performance comparison with the success rate of reaching goal positions during evaluation in an extended temporal configuration of the U-shaped and maze-shaped Ant navigation tasks. We find that integrating DGRL with RIS can lead to more sample efficient convergence on these tasks, while baselines such as SAC and TDM (not shown) fail completely on both AntU and AntMaze as reported by [7]. The RIS baseline is based on raw data provided by the authors.

**Vision Based Robotic Manipulation.** Finally, we assess DGRL on a hard vision-based robotic manipulation task, and use the same setup as in Section 6 to integrate DGRL with the state-of-the-art RIS baseline on the Sawyer task in Figure 10. This manipulation task is adapted from [42], where the baseline RIS is already shown to be superior to previous goal conditioning methods. The task of the agent is to control a 2-DoF robotic arm from image input and move a puck positioned on the table. The Sawyer task is designed for training and generalization. At test time, it evaluates the agent's success at placing the puck in desired positions in a temporally extended configuration. This is a challenging vision-based complex motor task, since test time generalization requires temporally extended reasoning. Results in Figure 10 show that DGRL improves the sample efficiency over the RIS baseline. Details of the experimental setup are provided in Appendix C.7.

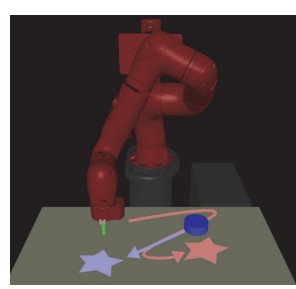 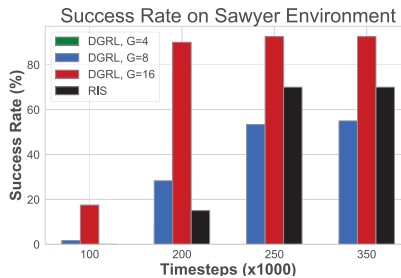

Figure 10: Sawyer Robotic Manipulation Task. Integrating DGRL with RIS (with a larger number of factors, G=8 and G=16, while G=4 fails), improves over the RIS baseline

# 7 Discussion

**Conclusion**   Our work provides direct evidence that performance of goal-conditioned RL can be improved when the representations of the goals are both *discrete* and *factorial*. We show that an instantiation of this idea using multi-factor discretization significantly improves performance on a diverse set of benchmarks.

**Limitations and Future Work**   An interesting question that arises from our work is how to theoretically *ground* and *specify* goals, which might be helpful for efficient structured exploration in tasks where goal seeking is crucial. Additionally, while we demonstrate that the factorial representations learnt by DGRL can be beneficial, it would be interesting to explore whether we can also enforce *compositionality* in the latent embeddings. Finally, exploring in more details the structure of the discrete factors forming the goals, their coverage, and to which extent they semantically capture the underlying factors of the environment is a promising research avenue.

# Acknowledgement

The authors would like to thank Nicolas Heess, John Langford, Yonathan Efroni, Manan Tomar, Dipendra Misra, Akshay Krishnamurthy, Pierre-Yves Oudeyer, Harm Van Seijen and Doina Precup for valuable discussions and insightful comments related to this work. Hongyu Zang and Xin Li were partially supported by NSFC under Grant 62276024.

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
