# A    Appendix

In section 5, we discuss extended related work. In section C we include details of our experiment setup and additional ablation analysis evaluating DGRL. Section D contains detailed proof for theorem 1. Finally, we provide our algorithm, DGRL that requires minimal changes to existing goal conditioned RL setup, in section E, with sample code snippet provided in F

# B    Demonstrating Factorized Representations in a Robot Experiment with Visual Background Distractors

We demonstrate the ability of DGRL to learn factorial representations on real world robot data, where the robot arm moves in presence of background video distractors [28]. Further details on the robot arm data collection are provided below. The data contains rich temporal background noise. We first learn a representation with a simple auto-encoder, following by the discretization bottleneck of DGRL, and then reconstruct the image with different discrete factors. Figure 11 demonstrates factorization in the learnt representation tweaking the different factors used in the discretization bottleneck. In particular, we observe some form of "compositionality" emerging as the decoder was never trained on some of the combinations of factors, for instance (person in the background + orange lamp) and (person in the background + arm to the left).

**Experiment Details:** In this task, the robot arm moves on top of a grid layout, containing 9 different positions. We denote these as the *true states*. We use two cameras to take images, for the dataset, one from the front side of the robot and the other with a top down view from above. We collect a dataset containing pixel based observations only, where the images consist of the robot arm along with the background distractors. Inspired by the exogenous noise information setup [**?** ], we setup the robot task while there is a TV playing a video in the background, with other flashing lights nearby. The offline dataset consists of 6 hours of robot data, with 14000 samples from the arm, taking high level actions of move left, right, up and down. A sample point image is collected after each action, and the background distractors changes significantly, due to video and lighting in the background. The goal of the experiment is to predict accurately the ground truth state position by learning latent representations with DGRL.

**Experiment Results:** We evaluate the ability of DGRL to accurately reconstruct the image, by learning the latent state representation while also ignoring the background distractors. This is denoted as the *Image Noise*, where we compare DGRL with and without VIB, alongside a baseline agent which only learns a representation. For learning latent representations, we use a multi-step inverse dynamics model [28]. In addition, we compare the ability of DGRL to accurately predict the ground truth states, denoted by *State Accuracy* solely from the observations, as a classification task. This is challenging since the learnt representation needs to predict ground states while ignoring the irrelevant background information. Furthermore with the learnt model, we predict the time-step for each observation as an additional metric to determine effectiveness of DGRL. The time-step is an indicator of the background noise that appeared in each sample; and with *Temporal Noise*, we evaluate DGRL to predict the time step while ignoring irrelevant information from observations. Experiment results in Figure **??** shows that the use of VIB helps improve the ability of DGRL to remove noise from the representation, while being able to almost perfectly predict the ground truth state of the robot.

# C    Experiment Details and Additional Results

For all our experiments, we use existing open-source implementations of the baseline algorithms. We mostly integrate DGRL with HRAC [72] and RIS [7], which are state of the art goal conditioned algorithms on the complex Ant navigation and robotic manipulation tasks. We use the same hyperparameters and default configurations as used in the baselines. For the simpler maze tasks such as from MiniGrid and KeyChest, we also use existing setups used by previous algorithms [6, 72]. All our experiment results are based on 3 random seeds, and we provide our implementation for reproducibility.

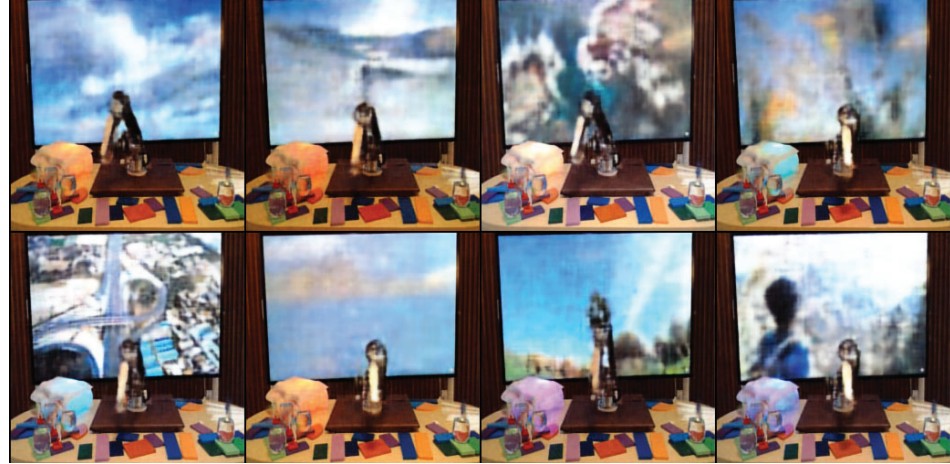

(a) Normal Reconstructions

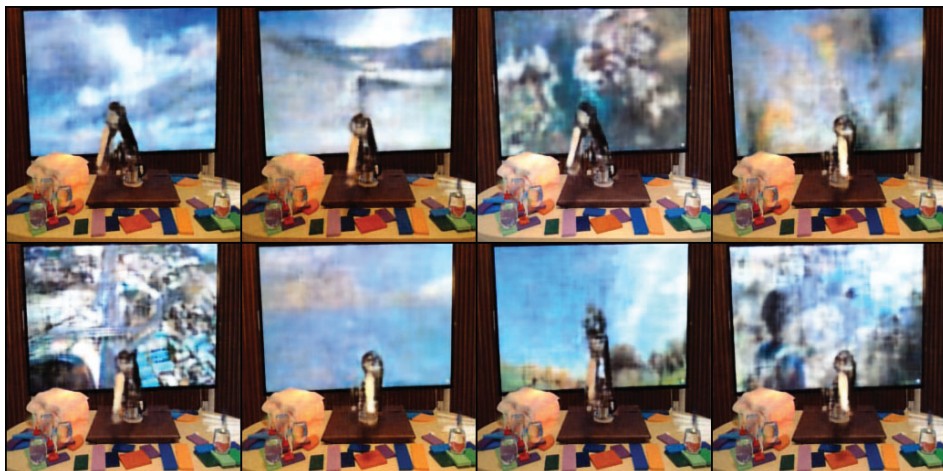

(b) Change Lamp to Orange (changing a single factor)

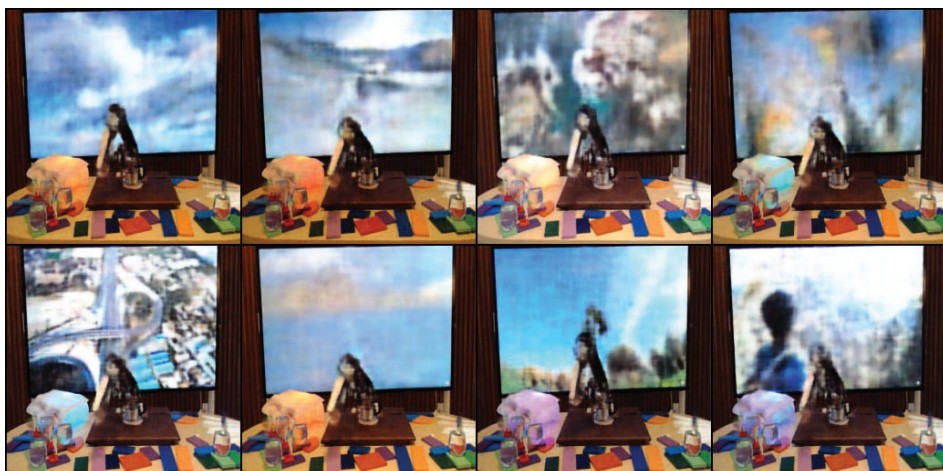

(c) Make arm point left (changing a single factor)

Figure 11: An autoencoder trained with 8 discrete codes to reconstruct images of a real-robotic arm with independently changing distractors (TV and lamp). The first two rows show the reconstruction of the initial images. In the middle two rows, we change one of the discrete factors to match its value in the first image, in the resulting reconstruction, the lamp has turned orange. On the bottom two rows, we change another discrete factor, to match its value from the first image, and observe that it made the robotic arm point left

### C.1 Experiment Details on Color-MNIST Dataset

We include a brief description of the experiment details, used for the color-mnist example to demonstrate factorization. The pixel-based input is first passed through an encoder (a two-layer neural network) to obtain its latent representation with the dimension of 30, we then quantize the continuous representation into two groups of discrete codes, where the codebook size is 256. In the training procedure, two groups of the discrete codes are then concatenated to obtain the discretized representation, and finally passed through a decoder (another two-layer neural network), where we used reconstruction loss (MSE loss) combining with the loss for vector quantization to train the network. While in the testing procedure, we used zero vector to substitute one group of the discrete codes, and then obtain the reconstructed image by concatenating it with the other group and passing through the decoder.

### C.2 Visual MazeWorlds

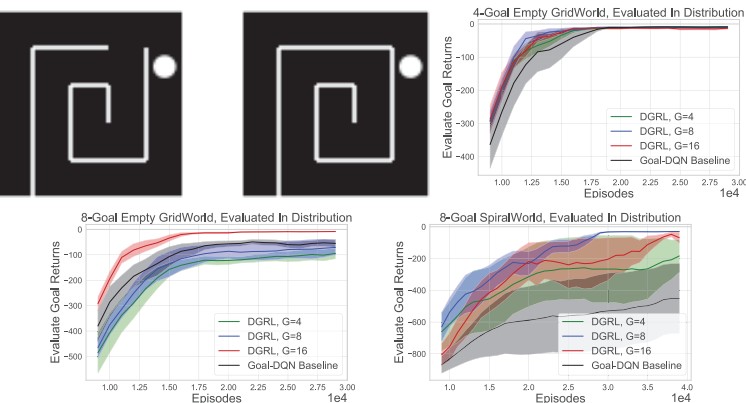

Figure 12: Visualization of the SpiralWorld and LoopWorld environments. Additional experimental results in an empty 4-goal and 8-goal environment across different factors $G$.

In this section, we provide additional details for the visual maze tasks. We provide environment visualizations, for the spiralworld and loopworld environments in figure 12. For this task, we use $6 \times 6$ gridworlds, where the agent receives pixel based full observations, of size $80 \times 80 \times 3$, of the environment. The agent starts deterministically in the environment from one corner of the loop, and the goals are provided in the state space, where agent receives observations of goals. The agent receives a negative reward of $-1$ at every time step, and a reward of $+5$ when reaching any of the goals.

We train a goal conditioned Q learning agent, with epsilon greedy exploration, where the agent receives goal observations in addition to state observations. We use a simple 2 layer architecture for the value function, which additionally conditions on the goals. In our experiments, we use 4-goals and 8-goals environment, where the agent is trained to reach each of the goals, sampled at every episode. For pixel based observations, we learn a representation $\phi$, which can be trained via any self-supervised representation learning objective. For our experiments, we use the Deep InfoMax algorithm [34] for training the representation learner $\phi$. Since the environment is quite simple, we found that even pre-training $\phi$ with random rollout policies is often good enough for training the value functions, as suggested by our experimental results. For DGRL, we additionally use a discrete bottleneck on top of the embedding, and experimentally show that DGRL can significantly outperform a baseline goal DQN agent.

### C.3 KeyChest Domain

For prototype and motivation of DGRL, we also used a simple discrete state and action KeyChest environment, as shown in figure 7. This task is inspired from the HRAC algorithm [72], where the agent starts stochastically in the environment, and the goal is to pick up a key and open a chest. We follow a HRL setup in this task, where the higher level policy provides discrete goals in the state

space for the lower level policy to reach. Since the environment has injected stochasticity, such a task requires both low-level control conditioned on the goals, provided by a higher level planner.

The environment has a 3-dimensional state space where the first two represent the position of the agent, while the third dimension represents whether the agent has picked up the key or not. The reward function is sparse, such that the agent only receives a reward $+1$ for picking up the key, and a reward of $+5$ if it can open the chest; otherwise zero rewards elsewhere. This hints to a hard exploration task, where specification of goal plays a key role. The lower level policy receives a goal reaching reward only, upon reaching the goal state. The higher level policy receives rewards from the environment directly. For the goal reaching reward, we use a standard Euclidean distance between states and goals, and the goal is achieved by the lower level policy if this distance is below a threshold of $0.5$.

For DGRL, we apply a discrete bottleneck by first learning a representation of the discrete goal in state space, using an encoder. We then apply a discrete bottleneck on the learnt embedding, followed by a decoder that maps the learnt discrete goal embedding back to the original state space. For baseline [72], the algorithm uses the raw goal states provided by the higher level policy. Both the higher and lower level policies are learnt with actor-critic algorithms. The high level policy is trained based on the task reward, whereas the lower level policy is rewarded for reaching the goals, or nearby regions of the goals, provided by the higher level policy.

## C.4    MiniGrid Environments

For our experiments, we follow the setup of [6] and [46] and evaluate DGRL on a simple procedurally generated MiniGrid environment [10]. The minigrid environments are a suite of hard exploration testbeds in RL where the task is designed such that exploration and representation of the visual observations can be disentangled. Following [6], we use the **KCharder**, KeyCorrS4R3 environment, which requires finding a key that can unlock a door which blocks a room. This door needs to be opened by the agent so as to reach the goal. In our experiments, we use the same learning rates, network architecture and other hyperparameters as in the AMIGO paper [6]. We use the open-sourced implementation provided by the authors (for more details, see [6]), and simply integrate DGRL on top of the AMIGO baseline, learning a factorial representation for the goal observations.

## C.5    Ant Manipulation tasks

We employ an ant robot with a continuous 8-dimensional action space for all three Ant manipulation tasks. Each episode terminates at 500 time steps in all three tasks.

**AntMazeSparse**    AntMazeSparse is a challenging navigation task with sparse rewards. This environment has a continuous state space including current position, velocity, target location and the current timestep $t$. The agent is provided with a sparse reward by $+1$ only if the Euclidean distance between the agent and the target location is smaller than 1, where the target location is set at $(2.0, 9.0)$ in the center corridor.

**AntPush**    This environment has the same state space as AntMazeSparse task. The difference is that this environment has a movable block which the agent can interact with. To successfully reach the target position $(0.0, 19.0)$, the agent must push the large block to the side to clear the path to the target location. The success of the agent is defined as having the Euclidean distance of 5 from the target position.

**AntFall**    This environment extends the navigation to three dimensions. Similar to the AntPush task, the environment still has a movable block, while the agent must move the block into a chasm instead of pushing it aside, so that it may walk over it without falling to reach the target position. The target position is fixed to $(0.0, 27.0, 4.5)$ in this environment.

For our experiments, we use the setup provided by the HRAC baseline [72]. In HRAC, an additional adjacency constrained is trained, along with the lower and higher level policies, such that the goals provided by higher controller are within a constrained region of the state space, that can be reachable by the lower level policy. The higher and lower level policies are both trained based on actor-critic

algorithms, with separate replay buffers for each policy. The replay buffer for the higher level policy stores one every K transitions.

We integrate DGRL on top of the HRAC baseline setup, where we apply the discrete goal bottleneck based on the output embeddings from higher level policies. Compared to HRAC, DGRL with HRAC would condition the lower level policies on the discrete embeddings of goals.

Figure 13 provides results for all the Ant manipulation tasks. Here we provide the results obtained for all the 5 different Ant environments, and we experiment for a range of discrete factors from 2 to 32. We find that while the best performing factor $G$ is not consistent for all tasks, in general factors of 4, 8 and 16 typically outperform 2 and 32 factors.

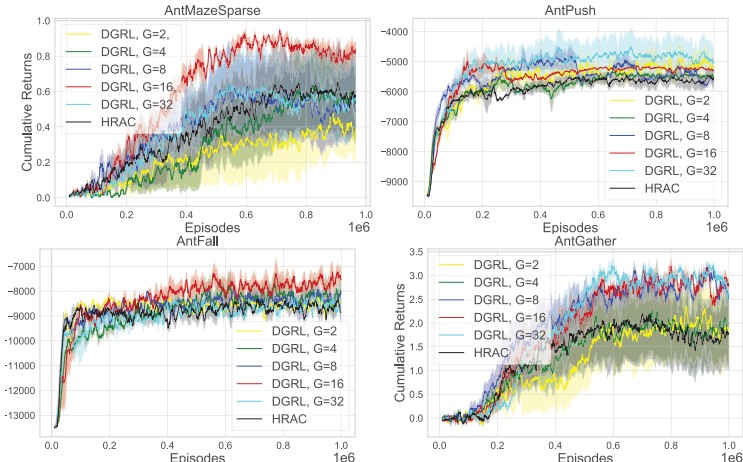

Figure 13: Ablation analysis on the Ant tasks. Standard goal conditioned HRL tasks, AntGather (shown on bottom left) and AntMaze, showing performance comparison of DGRL with baseline HRAC [72] across 3 random environment seed. We include comparisons with all factors $G$ for all the Ant manipulation environments that we expeirmented on, given computation budget.

## C.6 Ant Navigation Tasks

Following the setup in RIS [7], we employ an ant robot with a continuous 8-dimensional action space for all four Ant navigation tasks. reasoning. Figure 14(c) shows an additional result on the U-shaped Ant maze navigation environment. All these environments have a continuous state space including the current position, orientation, the joint angles and the velocities. The goal is considered reached when the Euclidean distance from the target position of the environment is less than 0.5. The agent gets a $-1$ reward at each time step until the goal is reached. The $U$-shaped maze has a size of $7.5 \times 18$, the $S$-shaped maze has a size of $12 \times 12$, while the $\pi$-shaped maze and $\omega$-shaped maze share the same size of $16 \times 16$. In the training stage, target and initial position of the agent are sampled randomly at the beginning of each episode. At evaluation, the initial state and the target position are fixed, as illustrated in Figure 14, to test the performance of the agent on challenging configurations that require temporally extended

## C.7 Robotic Manipulation Task

We consider the visual robot manipulation task Sawyer, from the multiworld environments of [42]. Our setup is based entirely on the experiment details and code provided in the open-source codebase of RIS [7]. The goal of the agent is to operate a 2D position control and manipulation task. The observations of the environment are based on $84 \times 84$ RGB image of the environment. DGRL is trained using a discrete bottleneck based on the learnt representation of the pixel based observations of the environment. We compare the Sawyer environment based on an existing RIS [7] baseline which is already shown to outperform other baselines on this task. For more details on the suite of multiworld environments, including the Sawyer manipulation task, see [42]. In this setting, we can additionally check for generalization. At test time, the cumulative returns of the agent are provided

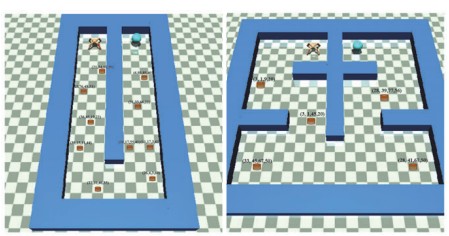
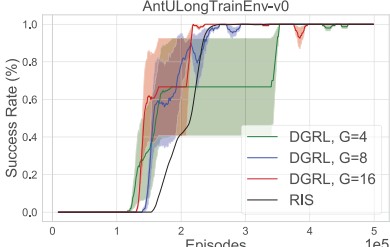

(a) U-shaped maze   (b) Π-shaped maze   (c) Experiment results comparing different factors $G$ for DGRL integrated with RIS, and compared with a RIS baseline (baseline data provided by authors)

Figure 14: Visualization of Ant navigation environment, which are standard goal conditioned HRL tasks. We integrate DGRL on top of the existing state of the art RIS baseline [7]. Performance comparison of success rate on AntU-shaped environment (right). We compare DGRL with different factors $G$ and the RIS baseline [7] which has already been shown to outperform other goal conditioned baselines such as LEAP [42] on the Ant navigation tasks.

when evaluated on a slightly different task, such as a hard configuration where we evaluate the policy and bottleneck based on learnt representations in the given task.

## D  Proof of Theorem 1

We consider a goal-conditioned Markov decision process, defined by states $s \in \mathcal{S}$, goals $g \in \mathcal{G}$, actions $a \in \mathcal{A}$, a reward function $r(s, a, g)$, a transition dynamics $p(s'|s, a)$, a maximum horizon $T$, the initial state distribution $\rho_0$, and the goal distribution $\rho_g$. The objective in goal-conditioned RL is to obtain a policy $\pi(a|s, g)$ to maximize the expected sum of rewards $\mathbb{E}_{g \sim \rho_g, ((s_t, a_t))_{t=1}^T}[\sum_{t=1}^T r(s_t, a_t, g)]$ where the sequence of state-action pairs $((s_t, a_t))_{t=1}^T$ is sampled according to $s_0 \sim \rho_0$, $a_t \sim \pi(a_t|s_t, g)$, and $s_{t+1} \sim p(s_{t+1}|s_t, a_t)$. To study the phenomenon of the goal observations, we define

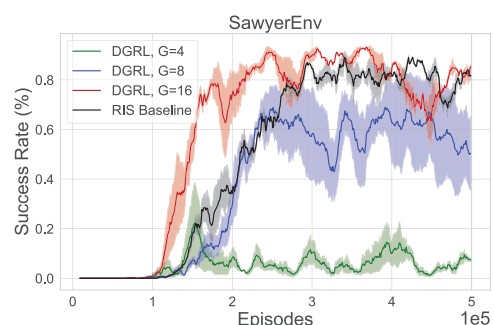

Figure 15: Performance comparison of DGRL with RIS baseline on a complex vision-based robot manipulation task

$$\varphi_\theta(g) = \mathbb{E}_{((s_t, a_t))_{t=1}^T}\left[\sum_{t=1}^T r(s_t, a_t, g)\right],$$

where $\theta \in \mathbb{R}^m$ is the vector containing model parameters learned through goals observed during training phase, $g_1, \ldots, g_n$. Define $\mathbb{1}\{a = b\} = 1$ if $a = b$ and $\mathbb{1}\{a = b\} = 0$ if $a \neq b$. Moreover, $\varphi_\theta \circ \varsigma$ represents the composition of functions $\varphi_\theta$ and $\varsigma$.

In our proof, we will use the following previous result:

**Lemma 1** (Bretagnolle-Huber-Carol inequality). *[65, Proposition A.6.6] If $X_1, \ldots, X_K$ are multinomially distributed with parameters $m$ and $p_1, \ldots, p_K$, then for $\bar{M} > 0$,*

$$\mathbb{P}\left(\sum_{k=1}^K \left|\frac{X_k}{m} - p_k\right| \geq \bar{M}\right) \leq 2^K \exp\left(-\frac{m\bar{M}^2}{2}\right).$$

*Proof.* We decompose the goal probability $p(g)$ into $p(g) = \sum_k p(g|g \in \mathcal{G}_k)p(g \in \mathcal{G}_k)$ with the neighborhood set $\{\mathcal{G}_k\}_k$ and analyze the concentrations of random variables in terms of $p(g|g \in \mathcal{G}_k)$

and $p(g \in \mathcal{G}_k)$. Let $\varsigma \in \{\mathrm{id}, q\}$. We define $M = \sup_{g \in \mathcal{G}} \varphi_\theta(g)$. Then,

$$\mathbb{E}_{g \sim \rho_g}[(\varphi_\theta \circ \varsigma)(g))] = \sum_k \mathbb{E}_{g \sim \rho_g}[(\varphi_\theta \circ \varsigma)(g)|g \in \mathcal{G}_k] \Pr(g \in \mathcal{G}_k),$$

Using this, we decompose the difference as

$$\frac{1}{n} \sum_{i=1}^{n} (\varphi_\theta \circ \varsigma)(g_i) - \mathbb{E}_{g \sim \rho_g}[(\varphi_\theta \circ \varsigma)(g))] \tag{2}$$

$$= \sum_k \mathbb{E}_{g \sim \rho_g}[(\varphi_\theta \circ \varsigma)(g)|g \in \mathcal{G}_k] \left( \frac{|\mathcal{I}_k|}{n} - \Pr(g \in \mathcal{G}_k) \right)$$

$$+ \left( \frac{1}{n} \sum_{i=1}^{n} (\varphi_\theta \circ \varsigma)(g_i) - \sum_k \mathbb{E}_{g \sim \rho_g}[(\varphi_\theta \circ \varsigma)(g)|g \in \mathcal{G}_k] \frac{|\mathcal{I}_k|}{n} \right).$$

Since $\frac{1}{n} \sum_{i=1}^{n} (\varphi_\theta \circ \varsigma)(g_i) = \frac{1}{n} \sum_k \sum_{i \in \mathcal{I}_k} (\varphi_\theta \circ \varsigma)(g_i)$,

$$\frac{1}{n} \sum_{i=1}^{n} (\varphi_\theta \circ \varsigma)(g_i) - \sum_k \mathbb{E}_{g \sim \rho_g}[(\varphi_\theta \circ \varsigma)(g)|g \in \mathcal{G}_k] \frac{|\mathcal{I}_k|}{n}$$

$$= \frac{1}{n} \sum_k |\mathcal{I}_k| \left( \frac{1}{|\mathcal{I}_k|} \sum_{i \in \mathcal{I}_k} (\varphi_\theta \circ \varsigma)(g_i) - \mathbb{E}_{g \sim \rho_g}[(\varphi_\theta \circ \varsigma)(g)|g \in \mathcal{G}_k] \right).$$

Substituting these into equation equation 2 yields

$$\frac{1}{n} \sum_{i=1}^{n} (\varphi_\theta \circ \varsigma)(g_i) - \mathbb{E}_{g \sim \rho_g}[(\varphi_\theta \circ \varsigma)(g))] \tag{3}$$

$$= \sum_k \mathbb{E}_{g \sim \rho_g}[(\varphi_\theta \circ \varsigma)(g)|g \in \mathcal{G}_k] \left( \frac{|\mathcal{I}_k|}{n} - \Pr(g \in \mathcal{G}_k) \right)$$

$$+ \frac{1}{n} \sum_k |\mathcal{I}_k| \left( \frac{1}{|\mathcal{I}_k|} \sum_{i \in \mathcal{I}_k} (\varphi_\theta \circ \varsigma)(g_i) - \mathbb{E}_{g \sim \rho_g}[(\varphi_\theta \circ \varsigma)(g)|g \in \mathcal{G}_k] \right)$$

$$\leq M \sum_k \left| \frac{|\mathcal{I}_k|}{n} - \Pr(z \in \mathcal{G}_k) \right| + \frac{1}{n} \sum_k |\mathcal{I}_k| \left( \frac{1}{|\mathcal{I}_k|} \sum_{i \in \mathcal{I}_k} (\varphi_\theta \circ \varsigma)(g_i) - \mathbb{E}_{g \sim \rho_g}[(\varphi_\theta \circ \varsigma)(g)|g \in \mathcal{G}_k] \right)$$

By using Lemma 1 by setting $\delta = 2^K \exp\left(-\frac{m\bar{M}^2}{2}\right)$ and solving for $M$, we have that for any $\delta > 0$, with probability at least $1 - \delta$, $\sum_k \left| \frac{|\mathcal{I}_k|}{n} - \Pr(z \in \mathcal{G}_k) \right| \leq \sqrt{\frac{2 \ln(2^{|Q|}/\delta)}{n}} \leq \sqrt{\frac{2|Q| \ln(2/\delta)}{n}}$, where the last inequality follows from the fact that $1/\delta \leq 1/\delta^{|Q|}$ as $\delta \in (0, 1)$ and $|Q| \geq 1$. Here, notice that the term of $\sum_k \left| \frac{|\mathcal{I}_k|}{n} - \Pr(z \in \mathcal{G}_k^y) \right|$ does not depend on $\theta$. Moreover, note that for any $(f, h, M)$ such that $M > 0$ and $B \geq 0$ for all $X$, we have that $\mathbb{P}(f(X) \geq M) \geq \mathbb{P}(f(X) > M) \geq \mathbb{P}(Bf(X) + h(X) > BM + h(X))$, where the probability is with respect to the randomness of $X$. Thus, by combining this and equation equation 3, we have that for any $\delta > 0$, with probability at least $1 - \delta$, the following holds for all $\theta$,

$$\frac{1}{n} \sum_{i=1}^{n} (\varphi_\theta \circ \varsigma)(g_i) - \mathbb{E}_{g \sim \rho_g}[(\varphi_\theta \circ \varsigma)(g))] \tag{4}$$

$$\leq \frac{1}{n} \sum_{k=1}^{|Q|} |\mathcal{I}_k| \left( \frac{1}{|\mathcal{I}_k|} \sum_{i \in \mathcal{I}_k} (\varphi_\theta \circ \varsigma)(g_i) - \mathbb{E}_{g \sim \rho_g}[(\varphi_\theta \circ \varsigma)(g)|g \in \mathcal{G}_k] \right) + c\sqrt{\frac{2 \ln(2/\delta)}{n}}$$

$$= \frac{1}{n} \sum_{k \in \mathcal{I}_Q} |\mathcal{I}_k| \left( \frac{1}{|\mathcal{I}_k|} \sum_{i \in \mathcal{I}_k} (\varphi_\theta \circ \varsigma)(g_i) - \mathbb{E}_{g \sim \rho_g}[(\varphi_\theta \circ \varsigma)(g)|g \in \mathcal{G}_k] \right) + c\sqrt{\frac{2 \ln(2/\delta)}{n}}$$

If $\varsigma = \mathrm{id}$, then

$$\frac{1}{n}\sum_{k\in\mathcal{I}_{\mathcal{Q}}}|\mathcal{I}_k|\left(\frac{1}{|\mathcal{I}_k|}\sum_{i\in\mathcal{I}_k}(\varphi_\theta\circ\varsigma)(g_i)-\mathbb{E}_{g\sim\rho_g}[(\varphi_\theta\circ\varsigma)(g)|g\in\mathcal{G}_k]\right)$$

$$=\frac{1}{n}\sum_{k\in\mathcal{I}_{\mathcal{Q}}}|\mathcal{I}_k|\left(\frac{1}{|\mathcal{I}_k|}\sum_{i\in\mathcal{I}_k}\varphi_\theta(g_i)-\mathbb{E}_{g\sim\rho_g}[\varphi_\theta(g)|g\in\mathcal{G}_k]\right)=\omega(\theta).$$

If $\varsigma = q$, then

$$\frac{1}{n}\sum_{k\in\mathcal{I}_{\mathcal{Q}}}|\mathcal{I}_k|\left(\frac{1}{|\mathcal{I}_k|}\sum_{i\in\mathcal{I}_k}(\varphi_\theta\circ\varsigma)(g_i)-\mathbb{E}_{g\sim\rho_g}[(\varphi_\theta\circ\varsigma)(g)|g\in\mathcal{G}_k]\right)$$

$$=\frac{1}{n}\sum_{k\in\mathcal{I}_{\mathcal{Q}}}|\mathcal{I}_k|\left(\frac{1}{|\mathcal{I}_k|}\sum_{i\in\mathcal{I}_k}(\varphi_\theta\circ q)(g_i)-\mathbb{E}_{g\sim\rho_g}[(\varphi_\theta\circ q)(g)|g\in\mathcal{G}_k]\right)$$

$$=\frac{1}{n}\sum_{k\in\mathcal{I}_{\mathcal{Q}}}|\mathcal{I}_k|\left(\frac{1}{|\mathcal{I}_k|}\sum_{i\in\mathcal{I}_k}\varphi_\theta(\mathcal{Q}_k)-\varphi_\theta(\mathcal{Q}_k)\right)=0$$

Therefore, for any $\delta > 0$, with probability at least $1 - \delta$, the following holds for any $\theta \in \mathbb{R}^m$ and $\varsigma \in \{\mathrm{id}, q\}$:

$$\mathbb{E}_{g\sim\rho_g}[(\varphi_\theta\circ\varsigma)(g))]\geq\frac{1}{n}\sum_{i=1}^n(\varphi_\theta\circ\varsigma)(g_i)-c\sqrt{\frac{2\ln(2/\delta)}{n}}-\mathbb{1}\{\varsigma=\mathrm{id}\}\omega(\theta).$$

Define $u_n(S,\theta) = \max_{k\in\mathcal{I}_{\mathcal{Q}}}\frac{1}{|\mathcal{I}_k|}\sum_{i\in\mathcal{I}_k}\varphi_\theta(g_i) - \mathbb{E}_{g\sim\rho_g}[\varphi_\theta(g)|g \in \mathcal{G}_k]$. Then, since $\sum_{k\in\mathcal{I}_{\mathcal{Q}}}|\mathcal{I}_k|=n$,

$$\omega(\theta)\leq u_n(S,\theta)\frac{1}{n}\sum_{k\in\mathcal{I}_{\mathcal{Q}}}|\mathcal{I}_k|=u_n(S,\theta).$$

For each $k \in [|\mathcal{Q}|]$, if $p(g \in \mathcal{G}_k) = 0$, then the probability of the event of $|\mathcal{I}_k| \geq 1$ is zero. Thus, by taking union bounds, with probability one, for all $k \in \mathcal{I}_{\mathcal{Q}}$, $|\mathcal{I}_k| \to \infty$ as $n \to \infty$. Therefore, by using the uniform law of large numbers (Theorem 2 of [21]) and union bounds over $k \in \mathcal{I}_{\mathcal{Q}}$ (noticing that $|\mathcal{I}_{\mathcal{Q}}| \leq |\mathcal{Q}|$ is finite), we have that $\sup_{\theta\in\Theta}|u_n(S,\theta)|\xrightarrow{P}0$ when $n \to \infty$. Thus,

$$0\leq\sup_{\theta\in\Theta}|\omega(\theta)|\leq\sup_{\theta\in\Theta}|u_n(S,\theta)|\xrightarrow{P}0\quad\text{when}\quad n\to\infty.$$

$\square$

# E   Algorithm in Goal Conditioned RL

We present the entire algorithm of DGRL built on top of RIS in algoirithm 1.

---

**Algorithm 1: RIS with DGRL (changes to RIS in blue)**

---

1: Initialize replay buffer $D$
2: Initialize $Q_\phi$, $\pi_\theta$, $\pi_\psi^H$
3: **for** k = 1, 2, ... **do**
4:     Collect experience in $D$ using $\pi_\theta$ in the environment
5:     Sample batch $(s_t, a_t, r_t, s_{t+1}, g) \sim D$ with HER
6:     Sample batch of subgoal candidates $z_e \sim D$
7:     Update $Q_\phi$ using Policy Evaluation
8:     Update $\pi_\psi^H$ using High-Level Policy Improvement
9:     Output subgoal $z_e$ using $z_e \sim \pi_{\psi_{k+1}}^H(\cdot|s, g)$
10:     Output discrete goal embedding $z_q$ using Eq. 1 *(Discretization module)*
11:     Compute prior policy with discrete sub-goal embeddings

$$\pi_k^{\text{prior}}(a \mid s, g) := \mathbb{E}_{z_e \sim \pi^H(.|s,g)} \left[ \pi_{\theta_k'}(a \mid s, z_q) \right] \tag{5}$$

12:     Update $\pi_\theta$ using Policy Improvement with Imagined Subgoals (Eq. 9 in RIS [7])
13:     Update discretization module using $\mathcal{L}_{\text{discretization}} = \frac{\beta}{G} \sum_i^G ||c_i - \text{sg}(e_{o_i})||_2^2$
14: **end for**

---

## F    Code Snippet of DGRL

We show a simple code snippet of algorithm 1 below.

```python
class RIS(object):
    def __init__(self):
        discrete_cfg = {'groups': self.args.groups, 'n_embed': self.
    args.n_embed}
        self.vq_layer = VectorQuantizerEMA(state_dim, discrete_cfg['
    n_embed'], discrete_cfg['groups']).to(device)
        params = list(self.critic.parameters()) + list(self.vq_layer.
    parameters())
        self.critic_optimizer = torch.optim.Adam(params, lr=q_lr)

    def select_action(self, state, goal):
        with torch.no_grad():
            state = self.encoder(state)
            goal = self.encoder(goal)
            goal_embed, _, goal_ind = self.vq_layer(goal)
            action, _, _ = self.actor.sample(state, goal_embed)
        return action

    def train(self, state, action, reward, next_state, done, goal,
    subgoal):
        if self.image_env: ## FOR SAWYER ENVS
            state_z = self.encoder(state)
            next_state_z = self.encoder(next_state)
            goal_z = self.encoder(goal)
            subgoal_z = self.mlp_encoder(subgoal)
            encoder_loss = self.driml_loss(state, next_state, action)
            if self.args.use_vq == 'true':
                _, vq_loss_state, _ = self.vq_layer(state_z)
                _, vq_loss_next_state, _ = self.vq_layer(next_state_z)
                _, vq_loss_goal, _ = self.vq_layer(goal_z)
                _, vq_loss_subgoal, _ = self.vq_layer(subgoal_z)
                rep_loss = vq_loss_state + vq_loss_next_state +
    vq_loss_goal + vq_loss_subgoal
            rep_loss += encoder_loss

        else: ## for non-image or non-pixel based envs - ANT ENVS
            state_z = self.mlp_encoder(state)
            next_state_z = self.mlp_encoder(next_state)
            goal_z = self.mlp_encoder(goal)
            subgoal_z = self.mlp_encoder(subgoal)
            encoder_loss = self.autoencoder_loss(state)
            if self.args.use_vq == 'true':
                _, vq_loss_state, _ = self.vq_layer(state_z)
                _, vq_loss_next_state, _ = self.vq_layer(next_state_z)
                _, vq_loss_goal, _ = self.vq_layer(goal_z)
                _, vq_loss_subgoal, _ = self.vq_layer(subgoal_z)
                rep_loss = vq_loss_state + vq_loss_next_state +
    vq_loss_goal + vq_loss_subgoal
            rep_loss +=encoder_loss
```