# OpenReview forum: "Discrete Compositional Representations as an Abstraction for Goal Conditioned Reinforcement Learning"
_NeurIPS.cc/2022/Conference — NeurIPS 2022 Accept_

### Official Review · Reviewer_h8LS · 2022-07-04

**Rating:** 6
**Confidence:** 3
**Soundness:** 3 good
**Presentation:** 3 good
**Contribution:** 3 good

**Summary:**

The paper presents an approach to improve the performance of learned goal-conditioned policies, by discretizing the goals using Vector Quantization. They call their framework Discrete Goal-Conditioned Reinforcement Learning (DGRL), and it consists of 3 stages.
1) learn representations of the visual input using self-supervised learning, using either an autoencoder or an InfoMax objective.
2) Process the continuous latent representations via a discrete codebook (generalisation of VQ-VAE to multiple codes) to obtain a discretized code.
3) Use the pre-trained model on RL tasks, either by giving discrete goals and maximizing an intrinsic reward which is a function of the similarity between the given goal and the discretized embedding of the current observation, or in a HRL setting, where a high level controller is tasked with outputting goals, which are then discretized and used to condition the low level policy.

The authors provide a theorem that the goal discretization improves the lower bound of the expected sum of rewards of unseen goals. They then provide a series of results comparing their method to a number of different baselines and different task domains. The task domains they chose were a number of navigation tasks in a gridworld environment with pixel observations, a set of exploration tasks in the MiniGrid environment, a maze navigation task in 2D, ant manipulation control and navigation domains, and the Sawyer Robotic Manipulation Task. In the majority of the presented tasks, variants of DGRL have shown to provide some benefit in either sample efficiency or higher final performance.


**Questions:**

In many places in the manuscript, the discrete representations learned by the codebook are characterized as compositional. I would argue that compositionality in the representations learned using neural networks, though desirable, is something we have not yet achieved as a community. In my opinion, the reason for this is that compositionality requires some kind of syntax in the representation, which needs to be interpreted by something else in a very precise and consistent way. To claim that a representation is compositional, one would have to show that consistent replacements of parts of the representations lead to consistent results in behaviour. For example, exchanging the $i$th discrete component with a specific token from the cookbook would cause the conditioned policy to always go to red objects, whereas a different component could correspond to the type of object (ball, cube etc) and yet another could correspond to a particular area of the maze. Moreover, a truly compositional representation would require that any given combination of these tokens would lead to the expected behaviour, and perhaps even require evidence of nested syntax. Due to the lack of such evidence, I would encourage the authors to remove the term “compositional” and instead use, perhaps, “factored” or “structured”. I would also be interested to hear the authors’ thoughts about my concerns.

**Limitations:**

I haven’t seen any critical assessment of limitations of the method, and would be very interested to hear the authors’ views on this. For example, how many different goals can we express in this method? How easily can this scale? How different are the resulting discrete codes between them?

**Strengths And Weaknesses:**

### Strengths

1. The method is simple to implement in existing architectures and uses established approaches that have been developed for some time now.
2. The authors did a comprehensive empirical study in many different domains and tasks. It is very useful to see how the method performs in all these domains.
3. The paper is overall very well written, and the ideas and results are communicated clearly.

### Weaknesses

The method does not provide a significant improvement in most environments, nor I consider to be necessarily original (see [1,2] for examples using a very similar idea). It is of course interesting to know the benefits of this addition, but as the results show, the results are comparable for the majority of tasks, except in some cases of gridworld or ant maze navigation and the sawyer arm tasks, where the DGRL seems to help with sample efficiency. For example, in Table 1, it is clear that some choices of G can even considerably harm performance compared to the baseline (see G=8). To me, this is not necessarily a reason to reject, as this idea is something that many others will think to try at some point. Publishing genuine empirical results showing the realistic improvements that should be expected by a simple addition to the method, such as the one presented in this work, should be encouraged in my opinion.
In terms of presentation, I found it a bit odd that the related work was in the appendix. Perhaps a bit part of the introduction could have been removed or delegated to the appendix, as it refers to high level ideas about goal grounding and goal specification which are not directly addressed by the paper.
 I have a comment about the use of the term “compositionality” in the paper used to refer to the discrete representation, which I elaborate more on in the “Questions” section below. I would urge the authors to remove the term compositional from the title as well. Please see the questions section below.

[1] Ozair, S., Li, Y., Razavi, A., Antonoglou, I., Van Den Oord, A. and Vinyals, O., 2021, July. Vector quantized models for planning. In International Conference on Machine Learning (pp. 8302-8313). PMLR.

[2] Fernández, F., Borrajo, D. (2000). VQQL. Applying Vector Quantization to Reinforcement Learning. In: Veloso, M., Pagello, E., Kitano, H. (eds) RoboCup-99: Robot Soccer World Cup III. RoboCup 1999. Lecture Notes in Computer Science(), vol 1856. Springer, Berlin, Heidelberg. https://doi.org/10.1007/3-540-45327-X_24

---

> ### Author Response · Authors · 2022-08-02
> **Thank you for detailed feedback and comments : Use of Term Compositionality**
>
> We thank the reviewer for detailed feedback on the manuscript.
>
> As per the comments, we have indeed updated the draft and included additional experimental results to demonstrate the factorial (previously termed compositional) representations that DGRL learns. Please see the generic response made to all reviewers to address this.
>
> We hope to have addressed the issues with the term "compositionality" through our range of additional experiments in the updated manuscript. We do note that going further, we would re-phrase this as "factorial" representations, and have demonstrated additional results  showing how this arises, with additional details on the experiment setup provided as well.

---

> ### Author Response · Authors · 2022-08-02
> **Improvement over baselines in most environments**
>
> Q1 : The method does not provide a significant improvement in most environments. It is of course interesting to know the benefits of this addition, but as the results show, the results are comparable for the majority of tasks, except in some cases of gridworld or ant maze navigation and the sawyer arm tasks, where the DGRL seems to help with sample efficiency.
>
> We have included an additional result, in figure 19 of the updated manuscript. Please refer to this result.
>
> We emphasize the significance of DGRL through this result comparison. The results we show in the paper were being transparent about the effects of the different groups of factors; which is why in our plots, we included comparisons with all groups $G$. Previously, we reported results without fine-tuning the group factor G for a fair comparison. In this figure, we show the best performing factor $G$ for each environment, that we can specifically fine-tune for the task; and as shown, we see that by simply adding DGRL on top of the existing HRAC DGRL can become more apparent.

---

> ### Author Response · Authors · 2022-08-08
> **Feedback on Updated Manuscript, Additional Results and Response to your Questions**
>
> Dear Reviewer,
>
> Thanks again for taking time to review our paper and providing feedback. We hope to have addressed all your questions and concerns. Please see the generic response to all reviewers, as well as the updated supplementary (appendix) containing additional results that may help improve understanding of the paper.
>
> We felt that your review was very helpful and instructive, so it would be very helpful to get feedback on our changes. We hope this can help re-evaluate the rating of our work, if applicable. We would also be grateful if you can let us know if there are any other concerns which we can address further.
>
> Thank you for your help and time.

---

> ### Comment · Reviewer_h8LS · 2022-08-08
> **Response to authors**
>
> I would like to thank the authors for their detailed response to most of my points and for the hard work they've put into updating the paper. I have amended my review scores accordingly.

---

> > ### Author Response · Authors · 2022-08-08
> > **Thank you!**
> >
> > Dear Reviewer,
> >
> > Thank you for updating the rating for the paper. Your feedback and commends have definitely helped improve the paper quite a lot. We appreciate your detailed review on our work.
> >
> > If there are any further questions or concerns, please let us know. We would try our best to address your questions for further improvement of our work.
> >
> >
> > Thanks.

---

### Official Review · Reviewer_PnN7 · 2022-07-06

**Rating:** 6
**Confidence:** 4
**Soundness:** 3 good
**Presentation:** 3 good
**Contribution:** 3 good

**Summary:**

This paper proposes to use discretized self-supervised representations for goal encoding in goal-conditioned RL. Specifically, the paper proposes using a codebook of L discrete codes to encode each of G chunks of a state/goal representation vector. The paper evaluates this method on a variety of environments for goal-conditioned and hierarchical RL, and demonstrates improved learning efficiency consistently, and improved generalization in several cases.


**Questions:**

See above, but briefly:
* Would this method perform as well if the space of possible goals were itself high-dimensional, rather than effectively 2D?
* How does the discretization change the training distribution, and can this be disentangled from the representation benefits?
* What are the benefits of choices like updating the codewords over training with EMA, and could these be justified with ablation studies?
* What are the codebook sizes?

**Limitations:**

See above, but briefly:
* It's unclear whether discretized methods would perform well in more complex settings; this should be explored and/or discussed.

**Strengths And Weaknesses:**

Strengths:
* The challenge of representing goals for RL is important, and the paper does a good job of motivating the problem and the proposed solution.
* The solution is relatively simple to implement, and the breadth of established evaluation environments and algorithms suggests that this approach will be generalizable to other similar settings
* The baseline comparisons seem generally strong, as they are taken from established works in the area.

Weaknesses:
* Discretized representations—even compositional ones—do not always scale well to high-dimensional, complex spaces. While the paper emphasizes the fact that the goals and states come from noisy, high-dimensional sensory inputs, the underlying goal spaces in each case are effectively 2D (goal positions to navigate to, to push a puck to, etc.), and thus it is unsurprising that a discrete representation would be effective. The strength of the paper would therefore be improved by also evaluating performance on higher dimensional goal spaces; for example, positioning 5 or 10 different objects in distinct positions with the robot rather than just 1, or stacking objects in particular places and orders, making sandwiches with different ingredients, etc. These experiments might be unreasonable to demand in the present submission—as the authors note, they evaluate on many environments from prior related works—but they would make the paper much stronger. Otherwise, it would be worth discussing this as a possible limitation explicitly.
* One aspect that seemed unclear in the current presentation is whether the goal reaching policy is trained according to the continuous goals produced by the higher-level policy or the discretized versions of them. For example, if the goal reaching policy is rewarded for getting within a Euclidean distance of 0.5 of the goal, is that based on the continuous or discrete representation? I assume the latter, but in that case the discretization process changes not only the way that the goal is represented to the agent, but the distribution of goals the agent is trained on. In particular, it seems like it might result in the goal reaching policy having a more consistent set of goals, which might explain some of the improved learning efficiency. These two factors could be disentangled in some ways, for example by discretizing the goal when inputting it to the agent, but still rewarding the agent based on the continuous representation. Exploring these issues would help to clarify the benefits of the method.
* Some more ablations could be useful. For example, what if the L latent vectors are fixed to reasonable values (e.g. the final codeword values of a prior training run) rather than updated with an exponential moving average? This might relate to the prior point, and would help identify whether something about the updating process is playing a role in the benefits, or whether it’s just the discretization.
* As far as I could find, the paper does not specify the codebook size L anywhere in the main text or the supplement. This detail is relevant to interpreting how discrete the encoding actually is, as well as for future researchers who want to use the method.
* To my understanding, text that is reproduced from a prior paper should be clearly marked as such; a non-trivial amount of section 3.2 is copied verbatim from one of the references without clearly communicating that fact (although the reference is cited as a source for the method). The paper should be transparent about details that are reproduced from prior work.
* A minor point, but the paper says “[26], which generalizes the vector quantization used in VQ-VAE [53] to a list of discretized codes instead of a single discretized code.”–the VQ-VAE paper also used arrays of codes rather than a single code in all applications; for example a grid of codes for images or an array of codes for sequences. It might be good to clarify this point.

---

> ### Author Response · Authors · 2022-08-02
> **Thank you for detailed feedback and comments**
>
> Thank you Reviewer oX1u for your detailed comments and questions. We try to address them accordingly. We answer each of the questions raised in details, and hope that it would provide further insights to the reviewer to re-evaluate their scores for this work.

---

> > ### Comment · Reviewer_PnN7 · 2022-08-04
> > **Thanks for your responses**
> >
> > Thanks to the authors for their responses; I have made one follow-up comment about the experiments and framing above.
> >
> > I do feel the paper has improved slightly with the edits, but I am not convinced that it has improved sufficiently to alter my rating. It seems still well described as "Technically solid, moderate-to-high impact paper, with no major concerns with respect to evaluation"

---

> ### Author Response · Authors · 2022-08-02
> **Goal space in Experiments and Limitations**
>
> Q1 : Goal space of current experiments and limitations
>
> Thank you for bringing this up. As we mentioned in the paper, our experimental results range from 2D goals spaces to even high dimensional image based goal spaces, and we show the effectiveness of DGRL across goal spaces of varying difficulty. As the reviewer pointed out, it would indeed have been interesting to do more experiments with high dimensional goal spaces, where for example the goals naturally have some level of factorization in the images, or goals consist of being able to interact with multiple objects. While interesting, and as the reviewer pointed out, these are indeed beyond the scope of our current work - not just due to computational or resource limitations, but we would like to point out that even existing goal conditioned RL benchmarks are not really designed (yet) to be adaptable to such settings. It definitely is an interesting future work that we would like to pursue, especially with the idea of learning multiple discrete goal representations which are factorial in nature. However, unfortunately, current approaches are beyond our capabilities, primarily also due to the fact that such baselines are hard to make it to work in practice without reliable existing setups.
>
> Reviewer oX1u also pointed out a similar approach - where for example goals can come from different modalities (e.g language based instructions as goals), and we emphasize that these are indeed interesting future works, and we hope DGRL provides a step towards that, due to the ability of capturing factorial representations. We hope that in future, goals that require interacting with objects of different colours and shapes, can perhaps be solved with extensions of DGRL, where truly understanding the causal and factorial representation learning capability is an interesting future work.

---

> ### Author Response · Authors · 2022-08-02
> **Continuous versus Discrete Goals**
>
> Q2 : Continuous vs Discrete Goals and what the goal reaching policy is trained on. For example discretizing the goal when inputting it to the agent, but still rewarding the agnet based on the continuous representation
>
> Please see figure 16 for an additional result in the manuscript.
>
> Following your suggestion, we ran an additional experiment on the KeyChest task, where we compare the performance of an intrinsic reward based on the discrete goal or on its continuous representation (while still passing a discrete goal to the agent). We see that the discrete version outperforms the continuous one.

---

> > ### Comment · Reviewer_PnN7 · 2022-08-04
> > **Thanks, given these results I think the framing is a little too focused on discrete goal representations rather than discrete goals**
> >
> > Thanks for running these experiments. I was most interested, in fact, in the comparison of the discrete rep + continuous goal to the baseline condition, so I appreciate you including that in the paper.
> >
> > My interpretation of the results would be that discrete goal representations per se are not helpful (in fact they are worse than baseline, as might be expected from the mismatch between the goals and the way they are presented to the agent). Given this, I feel that the framing focuses a bit too much on the discreteness of the goal representations per se, when the fact that the goals themselves are also discrete is also critical. For example, in the conclusions: "goal-conditioned RL can be improved when *the representations* of the goals are both discrete and factorial"—it seems to me a more appropriate statement would be that it can be improved when the curriculum goals are discrete *and* their representations are discrete and factorial.

---

> > > ### Author Response · Authors · 2022-08-05
> > > **Clarification of Discrete Goal vs Discrete Goal Representations**
> > >
> > > Thank you for your quick response.
> > >
> > > We would like to further clarify our additional experiment result, as it seems there might have been a mis-interpretation of the updated result.  According to figure 16 in the updated supplementary : let us first clarify that there are two components (a) discrete codes and the corresponding embeddings of the discrete codes (from VQ-VAE bottleneck) and continuous representations from baseline (b) intrinsic reward computed for the lower level policy based on the discrete code embeddings vs continuous embeddings.
> > >
> > > In figure 16, other than the HRAC baseline, we use discrete bottleneck in both; which is why the legend says "Discrete Goal"; however, in one of them, for computing the intrinsic reward, we use the embeddings of the discrete code (legend : Discrete Goal, Continuous Intrinsic) and in another, we use the discrete codes itself to compute the intrinsic reward (legend : Discrete Goal, Discrete Intrinsic).
> > >
> > > For all the methods, the state space is discrete and the higher level policy outputs (x,y) positions in the grid. We then learn an embedding of the 2D positions, followed by the discrete bottleneck - and change the way the lower level policy is trained (continuous intrinsic vs discrete intrinsic). For the discrete intrinsic, the reward is computed based on the frequency of the number of times the discrete code of the state matches the discrete code of the goal (since we are learning latent codes and latent embeddings of the codes for both the states and goals, when using DGRL).  Our experiment result shows that for this simple example, using the discrete code for the intrinsic reward is indeed useful, compared to using the embeddings of the discrete codes from VQ-VAE; and using the discrete latent codes itself for the reward computation can outperform the baseline.

---

> > > > ### Author Response · Authors · 2022-08-05
> > > > **Clarification : Experiments for both discrete goals and pixel/image based goal observations**
> > > >
> > > > We would like to clarify that for all our experiments, whether the goal is discrete or not is indeed not critical.
> > > >
> > > > We have experimented both with 2D discrete goals, and with image based goals (e.g in the Sawyer task, as in Figure 9 of main draft). In fact, figure 5 based on the MiniGrid environments also uses high dimensional goal observations.
> > > >
> > > > Our experiments covering the range of possible goal spaces was mainly to show that DGRL can perform well irrespective of what the underlying goal space is, in the benchmark tasks that we considered.
> > > >
> > > > We hope this clarifies the doubts from the reviewers. Please also see the reply titled "Goal space in Experiments and Limitations" for further clarification on this.

---

> > > > > ### Comment · Reviewer_PnN7 · 2022-08-08
> > > > > **Thanks for the clarification**
> > > > >
> > > > > Thanks for your clarification. I misunderstood the supplemental experiment you had performed, my apologies. However, I correspondingly find my original question in the review still unanswered, though it relates to the two components you highlight. I will try to restate my question in a way which may elucidate the issue.
> > > > >
> > > > > Imagine we are at a fixed step in the training process. I believe that at that step, the cardinality of the set of possible goals the agent can be tasked with reaching using HRAC differs from the cardinality of the set of possible goals using DGRL. In particular, DGRL results in a strictly smaller set of possible goals at any point in training. It's possible that this restriction helps the agent to learn, by making the goal sampling more consistent. Thus, the benefits of DGRL may be partly due to this difference, rather than the discreteness of the goal representations presented to the agent, which is emphasized in the paper.
> > > > >
> > > > > In order to assess whether this is contributing, it would be possible to run an experiment where the goals are sampled as per the HRAC baseline, the agent is rewarded as per the HRAC baseline, but *only* the goal embedding presented to the agent is discretized. That would be a clean demonstration of the benefits of discrete goal representations *per se*, rather than the additional changes introduced by DGRL. Or, if the agent would not perform as well there, then it suggests that the paper's emphasis should perhaps focus more on the contribution of other factors like the change in possible goal distribution.
> > > > >
> > > > > Please let me know if I'm still misunderstanding something, and thanks again for engaging with these comments.

---

> > > > > > ### Author Response · Authors · 2022-08-09
> > > > > > **We greatly appreciate your suggestion**
> > > > > >
> > > > > > Dear Reviewer,
> > > > > >
> > > > > > Thank you for your time in discussing this, and explaining it further with details. It has already done a lot to make the paper better.
> > > > > >
> > > > > > As we understand it, your question is about whether most of the improvement comes from the goal representations themselves being discretized, or from selecting a better (as you suggest, more selective) set of goals during training?
> > > > > >
> > > > > > We will do the suggested experiments, and add the results in the camera ready version of our paper.
> > > > > >
> > > > > >
> > > > > > Thanks.

---

> > > ### Author Response · Authors · 2022-08-05
> > > **Clarification if any other additional questions**
> > >
> > > Dear Reviewer,
> > >
> > > We would like to clarify if there are any other additional questions about our algorithm/method and experimental evaluation procedure.
> > >
> > > Given our work is purely experimental, we tried to cover a range of experiments, taking existing benchmark implementations from goal based RL literature. As such, due to different experiment settings, we agree it may see confusing about the different setups for each experiment. We tried to include as much details on the different setups in the appendix.
> > >
> > > We appreciate your detailed feedback and questions about our work, and look forward to answering more questions during the discussion period, which is definitely helping us to improve clarity in the manuscript.
> > >
> > >
> > > Thanks.

---

> ### Author Response · Authors · 2022-08-02
> **Q3 : Specifying the codebook size L**
>
> Indeed, it is missing from the manuscript, our apologies. For the toy tasks, we used a codebook size of $64$, while for the more complex control tasks, we used a codebook size of $128$, with different groups of factors ranging from $2, 4, 8, 16$ and $32$ (we did not optimize for L). With more compute available, it would be interesting to do a sweep on the codebook size $L$, the number of embeddings and the number of factors, to get a clearer picture of the impact of each on performance.

---

> ### Author Response · Authors · 2022-08-02
> **Goal Space : 2D Goals versus High Dimensional Goal Space**
>
> Q4 : Would this method perform as well if the space of possible goals were itself high-dimensional, rather than effectively 2D?
>
> As we mentioned earlier, our experimental results consist of standard goal conditioned RL benchmarks, and we do a wide range of experiments, starting from 2D goal spaces (toy tasks) to high dimensional or visual goal spaces (control manipulation tasks). We emphasize that in either case, given a representation objective, DGRL is a simple integration that can be applied to any existing self supervised or representation learning objective based method that involves learning an encoder (from small state spaces to pixel based input); and we hope that our contribution can be seen as a simple, yet effective methodology, based on discretization bottlenecks, that can be straightforwardly applied to RL tasks to significantly improve performance.

---

### Official Review · Reviewer_oX1u · 2022-07-11

**Rating:** 6
**Confidence:** 4
**Soundness:** 1 poor
**Presentation:** 2 fair
**Contribution:** 1 poor

**Summary:**

The paper proposes a learning framework for compositional representations of goals for goal-conditioned RL, and proposes to obtain a coarse specification of goals using discretization. The goals reside in a low-dimensional representation space that is obtained from high-dimensional sensory data,

**Questions:**

1. How are the training goals and new goals for evaluation in the MazeWorld environment being selected?
2. Does the method always require a dataset of goals collected prior to training, in order to train the goal representations? Do the curves reflect the sampling cost of these pre-collected data?
3. How does the method compared to end-to-end training, where goals and observations are encoded into a latent space and this encoder is learnt jointly with the policy?


**Limitations:**

The limitations and societal impact are not discussed. One limitation is that while a discretized goal representation enables composition, the composition is not necessarily interpretable, which is claimed in the introduction (line 78).

**Strengths And Weaknesses:**

Strengths
1. The paper shows favorable performance over several baselines on multiple environments.

Weaknesses
1. Why is it necessary to use two different autoencoder architecture for environments with different difficulty, if the author claims that representation learning method for observations can be used as described in Section 3.1?
2. The method assumes that the observations and goals come from the same space, restricting the method from being used when goals are specified in a different modality. This assumption should be discussed in the limitations.
3. The paper claims that the compositional representation helps encode goals into a semantically meaningful latent space (line 77), for example based on the example in Figure 1, I would expect some segments to correspond to goal colors and some segments to correspond to fruit types. However, HRL experiments in Section 5 do not provide evidence that the learnt goal representations are semantically interpretable.

---

> ### Author Response · Authors · 2022-08-02
> **Response to all technical questions and details**
>
> Thank you Reviewer oX1u for your detailed comments and questions. We answer each of the questions raised in details as below :
>
> Q1 : Why is it necessary to use two different autoencoder architecture for environments with different difficulty :
>
> Our DGRL approach is in fact robust to whether we use the self-supervised representation loss with discretization during a pre-training phase (as in the maze tasks) or end-to-end (as in the control tasks). In both cases, DGRL with a 16-factors in the discretization bottleneck can significantly improve empirical performance. Moreover, for our experiments, we built on existing codebases and results provided for the baselines, to ensure reproducibility and fair comparison we reused the code out of the box, including the encoder architecture. Our results show that DGRL is robust to architecture and task changes. The approach is a simple addition to existing baselines and provides gains across the board.

---

> ### Author Response · Authors · 2022-08-02
> **Compositional Goals and Semantically Meaningful Latent State Space**
>
> "Q3 : The paper claims that the compositional representation helps encode goals into a semantically meaningful latent space (line 77), for example based on the example in Figure 1, I would expect some segments to correspond to goal colors and some segments to correspond to fruit types. However, HRL experiments in Section 5 do not provide evidence that the learnt goal representations are semantically interpretable."
>
> Please refer to generic response to all reviewers, where we try to address this question in broader details.
>
> We emphasize that the traditional goal based RL benchmarks do not also consider such tasks where goals can be semantically interpretable. If such benchmarks were available to us, we could show reasonable results of how DGRL can be useful for semantic separability of goals, consisting of different colors and objects. For our experiments, we mostly considered the standard goal based tasks considered in the literature. However, that being said, in the rebuttal phase, we do add additional results for a wide range of image based data, either in a supervised learning setting or a RL setting, where we show how the multi-factor discretization considered in DGRL can capture different factors of variation in the input image. Please see updated draft where we provide demonstrations of the compositional or factorial representations captured by the model. We want to emphasize that in the updated version of the draft, as suggested by reviewer $h8LS$ , we do rephrase the term compostionality to be called factorial representations.

---

> ### Author Response · Authors · 2022-08-02
> **Observation and Goals, with goals specified by different modalities**
>
> Q2 : The method assumes that the observations and goals come from the same space, restricting the method from being used when goals are specified in a different modality.
>
> We acknowledge that this is a valid consideration, we indeed do not consider settings where goals can come from different input modalities (e.g goals provided by language instructions). We only considered standard goal based RL settings, but it is definitely an interesting avenue for future research.
>
> We also want to point that in some of our experiments, such as the SawyerTask or other control tasks - the goals are provided by visual images, part of the observation space. We tried to cover a wide range of task setups, starting from toy examples to several standard traditional goal based RL benchmarks, and also ranged from settings where goals are either provided by a higher level controller, or part of the environment itself. We believe the breadth of experiments considered to be convincing as to why DGRL can be a simple, but powerful addition to goal-based RL algorithms.

---

> ### Author Response · Authors · 2022-08-02
> **Training and Evaluation of Goal Selection**
>
> Q4 : How are the training goals and new goals for evaluation in the MazeWorld environment being selected?
>
> In the MazeWorld experiments, we designed the task such that there is a training distribution of goals (goals placed in different locations in the maze; typically 8 different goal placed in different positions during training). At the beginning of each episode, we sample a goal from the training distribution and pre-train a representation given these samples, followed by goal conditioned DQN agent training.
>
> During the evaluation phase, we test with both "within distribution" and "out of distribution" goals. In the "within distribution" setting, we evaluate the DQN agent on the same training set of goals and measure the cumulative returns for it to be able to reach these goals within the distribution. In the "out of distribution" setting, we explicitly choose the goals to be in different locations than the ones in the training set, and evaluate the agent to reach out of distribution goals.

---

> ### Author Response · Authors · 2022-08-02
> **Pre-Training of Goal Representations or End to End Training**
>
> Q6 : How does the method compared to end-to-end training, where goals and observations are encoded into a latent space and this encoder is learnt jointly with the policy?
>
> We apologize for the lack of clarity in this, and would update with more details in the next version.
>
> We emphasize here that we indeed consider both the settings, where for some simpler tasks we consider only training representations with DGRL during the pre-training phase; whereas in some of the more complex control tasks, we simultaneously or jointly learn the encoder and the policy and value functions (ie, end to end training). Our experimental results in the mazeworld environments show that adding DGRL in both the pre-training phase, as considered in task Loopworld and SpiralWorld environments or end to end training (joint training), as considered in the goal image based Sawyer tasks and image based Ant navigaton task.

---

> > ### Author Response · Authors · 2022-08-02
> > **Dataset of Pre-Collected Goals**
> >
> > Q5 : Does the method always require a dataset of goals collected prior to training, in order to train the goal representations? Do the curves reflect the sampling cost of these pre-collected data?
> >
> > Please refer to the answer above for Q6 as well. For our experiments, we consider both the settings where we train DGRL on pre-collected data (ie, where representations are training during a pre-training phase only), or a setting where goals are collected online by the sampling policy, and we jointly train the encoder and policy/value networks end to end.

---

> ### Author Response · Authors · 2022-08-08
> **Looking forward to your feedback, based on updated manuscript and response to all your questions**
>
> Dear Reviewer,
>
> Thanks again for taking time to review our paper and providing feedback. We have updated the paper, as well as added detailed replies to all of your comments (after conducting more experiments). We felt that your review was very helpful and instructive, so it would be very helpful to get feedback on our changes (and also update your rating if applicable).
>
> We would also be grateful if you can let us know if there are any other concerns which we can address further.
>
> Thanks for your help and time.

---

> > ### Comment · Reviewer_oX1u · 2022-08-08
> > **Thank you for the responses**
> >
> > I appreciate the clarifications and extra experiments. The clarifications on experimental designs to be consistent with baselines well addressed my concerns. The additional qualitative results that demonstrate the learnt factorization provide a glance over the learnt latent space. The authors also accommodated wordings such that the experiments presented match their claims. I've increased my rating accordingly.

---

> > > ### Author Response · Authors · 2022-08-09
> > > **Thank you for updating the scores**
> > >
> > > Dear Reviewer,
> > >
> > > Thank you very much for updating the scores for the paper. We appreciate your detailed feedback and comments on our work, which helped improve the paper significantly.
> > >
> > > If you have any further feedback on the draft, please let us know too.
> > >
> > >
> > > Thanks.

---

### Author Response · Authors · 2022-08-02
**Generic response to all reviewers : Updated draft with New Experiments demonstrating "Factorial" (or previously termed "Compositional") Representations**

We sincerely thank the reviewers for the insightful comments about the term compositionality used throughout the paper. We are taking account of this, and going forward, would re-phrase this term to mention DGRL learns factorial or factorized representations. The use of the term compositionality was based on some related experimental insights we had when working on this paper, which we omitted previously for sake of brevity and conciseness. We note that the use of the discretization bottleneck indeed provides a semantically meaningful and often indistinguishable latent space, depending on the number of factors.

For clarification to reviewers, we provide a set of new demonstrations to show how DGRL learns factorized representations. Additional details for the experiments are provided at the end of the manuscript (please see updated draft).

Please see Sections G and H in updated manuscript.

Experiment 1 : Color-MNIST Example for Factorization
We first give a supervised learning example based on the color-MNIST dataset, where factorization in the learnt latent space is achieved. Figure 14 in updated draft displays reconstructed images from a trained decoder operating on a discretized 2-factor representation. We see that different factors capture information of different semantic nature. More precisely, factor 1 tends to encode the shape of the digit, while factor 2 specialized in its color. This empirically demonstrates the emergence of ``factorization'' in the learnt representations.

Experiment 2 : Robot Experiment Visual Data with Background Distractors
We then demonstrate the ability of DGRL to learn factorial representations on real world robot data, where the robot arm moves in presence of background video distractors. The data contains rich temporal background noise. We first learn a representation with a simple auto-encoder, following by the discretization bottleneck of DGRL, and then reconstruct the image with different discrete factors. Figure 15 demonstrates factorization in the learnt representation tweaking the different factors used in the discretization bottleneck. In particular, we observe some form of ``compositionality'' emerging as the decoder was never trained on some of the combinations of factors, for instance (person in the background + orange lamp) and (person in the background + arm to the left).

---

> ### Author Response · Authors · 2022-08-02
> **Comments from Reviewers on "Compositional Representations" - Response with additional results in updated draft (sections G and H)**
>
> There were two comments from reviewers on the usage of the term compositionality, and what it means :
>
> Question From Reviewer oX1u : The paper claims that the compositional representation helps encode goals into a semantically meaningful latent space (line 77), for example based on the example in Figure 1, I would expect some segments to correspond to goal colors and some segments to correspond to fruit types.
>
>
> Comment from Reviewer h8LS : I have a comment about the use of the term “compositionality” in the paper used to refer to the discrete representation, which I elaborate more on in the “Questions” section below. I would urge the authors to remove the term compositional from the title as well. In many places in the manuscript, the discrete representations learned by the codebook are characterized as compositional. I would argue that compositionality in the representations learned using neural networks, though desirable, is something we have not yet achieved as a community. In my opinion, the reason for this is that compositionality requires some kind of syntax in the representation, which needs to be interpreted by something else in a very precise and consistent way. To claim that a representation is compositional, one would have to show that consistent replacements of parts of the representations lead to consistent results in behaviour.
>
> =======================
>
> We answer both these questions with additional experimental results demonstrating that DGRL can learn factorized representations.
>
> Please refer to Figures 14 and 15, demonstrating this, on few different tasks. We do acknowledge that the usage of this term may be mis-aligned, and have therefore updated the manuscript.
>
> We hope the additional results would provide new insights in understanding our method/approach, and would help reviewers to re-evaluate the scores.

---

> > ### Author Response · Authors · 2022-08-02
> > **Additional Experiment Details for Tasks used to demonstrate compositionality or factorization**
> >
> > Please refer to the updated draft, supplementary material, where at the end of appendix section, we included additional results with details for each task. We further repeat that here for clarity.
> >
> > Experiment Details on Color-MNIST Dataset :
> >
> > We include a brief description of the experiment details, used for the color-mnist example to demonstrate factorization. The pixel-based input is first passed through an encoder (a two-layer neural network) to obtain its latent representation with the dimension of 30, we then quantize the continuous representation into two groups of discrete codes, where the codebook size is 256. In the training procedure, two groups of the discrete codes are then concatenated to obtain the discretized representation, and finally passed through a decoder (another two-layer neural network), where we used reconstruction loss (MSE loss) combining with the loss for vector quantization to train the network. While in the testing procedure, we used zero vector to substitute one group of the discrete codes, and then obtain the reconstructed image by concatenating it with the other group and passing through the decoder.
> >
> > Experiment Setup on Robot Arm with Background Visual Distractor :
> >
> > The robot arm in our experiments moves in a grid with $9$ different positions. We use two cameras to take images, for the dataset, one from the front side of the robot and the other with a top down view from above. We collect an image after each action is taken. The robot has 5 actions to take : move forward, backwards, right, left or stay in the current state. We use an episodic length of $500$, ie, the robot arm moves for $500$ steps after which we re-calibrate. The robot arm dataset is collected with a random uniform policy, for a total of $6$ hours collecting $14000$ samples. For learning the representation $\phi$ given the images, we use a small convolutional neural network to get an estimate $\phi(x)$ of the images $x$. In addition to the CNN network, we further learnt the latent state representation with an autoencoder.

---

### Author Response · Authors · 2022-08-02
**Summary of Responses and Updated Manuscript**

We thank all the reviewers for their time and consideration into providing useful feedbacks for this manuscript. We hope our responses would help them re-evaluate their scores, as we tried to address all their concerns, both technical and high level comments.

We have updated our draft with new experiment results demonstrating how DGRL achieves factorized representations. This should hopefully provide better insights into why adding DGRL on the range of goal conditioned tasks that we considered experimentally, helps significantly improve performance.  Most importantly, we added new set of results (at the end of appendix, rebuttal section in draft) to show what kind of factorized representations DGRL achieves.

Furthermore, we have also addressed each of the reviewer questions individually as responses here, and have also included these responses in the updated draft (end of appendix, sections G and H).

We sincerely hope this would help reviewers understand this work better, and help them re-evaluate the scores for the paper.

---

### Author Response · Authors · 2022-08-09
**Summary of Discussion with Reviewers, Response to Questions and Our Changes**

Thank you for very thorough feedback. We are enthused by the positive feedback from all the reviewers, and excited that based on our discussion, the reviewers updated the rating of our work. The comments from reviewers helped improved the presentation of our work, and we have updated the appendix/supplementary with results based on reviewer feedback. We hope to incorporate all these changes in the final version of the manuscript.

Summary based on the discussion with reviewers.

We believe the most important update of our work  is based on the feedback on learning factorial representations. Our analysis, as in the updated supplementary/appendix of the paper, helps clarify further how DGRL learns factorial representations. We are delighted that the reviewers found the additional results to be extremely helpful for further clarification on this.

- **Compositional Representations and Semantically Meaningful Latent Space:** We have provided additional results in appendix to address reviewer concerns, and also changed our terminology to be "factorial" instead of "compositional". Please see Sections G and H in updated manuscript.

- **End to End Training versus Pre-Training of Goal Representations**:  We indeed considered both the settings, where for simple tasks, we can do both pre-training or end to end training; and in complex control tasks, we do end to end joint training.

- **Use of discrete goal representations versus discrete goals** : Based on reviewer questions on the use of continuous vs discrete goals and what the goal reaching policy is trained on - we included an additional result in figure 16 for comparison.

- **Goal space of current experiments and limitations:** Our experimental results range from 2D goals spaces to even high dimensional image based goal spaces. We believe the breadth of experiments considered to be convincing as to why DGRL can be a simple, but powerful addition to goal-based RL algorithms.


We would like to thank the reviewers again for engaging in a meaningful discussion. We will update our draft, with all the additional results/comments from the rebuttal phase, in the final version of the paper.

---

### Meta-Review · Area_Chair_r5rw · 2022-08-25

**Recommendation:** Accept
**Confidence:** Certain

**Metareview:**

This paper proposes a discrete and compositional representation of goal states for goal-conditioned RL. The idea is to learn a goal representation via self-supervised learning and discretize the learned representation via VQ-VAE, and finally use the learned goal representation for goal-conditioned RL. The proposed method improves performance on several goal-conditioned RL benchmarks.

All of the reviewers found the idea simple and reasonable, and the results on a variety of benchmarks are quite comprehensive and strong. Although there were concerns around why the proposed discretized representation forms a semantically meaningful latent space and where the improvement comes from, the authors addressed them during the rebuttal period with updated results. All of the reviewers became in favor of the paper as a result. Thus, I recommend accepting this paper.

**Award:**

No

---

### Decision · Program_Chairs · 2022-09-14

Accept